# Targeted viral adaptation generates a simian-tropic hepatitis B virus that infects marmoset cells

Yongzhen Liu [1], Thomas R. Cafiero[1], Debby Park[1], Abhishek Biswas[1,2], Benjamin Y. Winer[1,3], Cheul H. Cho[4], Yaron Bram[5], Vasuretha Chandar[5], Aoife K. O' Connell[6], Hans P. Gertje[6], Nicholas Crossland [6,7], Robert E. Schwartz [5] & Alexander Ploss [1] ✉

Hepatitis B virus (HBV) only infects humans and chimpanzees, posing major challenges for modeling HBV infection and chronic viral hepatitis. The major barrier in establishing HBV infection in non-human primates lies at incompatibilities between HBV and simian orthologues of the HBV receptor, sodium taurocholate co-transporting polypeptide (NTCP). Through mutagenesis analysis and screening among NTCP orthologues from Old World monkeys, New World monkeys and prosimians, we determined key residues responsible for viral binding and internalization, respectively and identified marmosets as a suitable candidate for HBV infection. Primary marmoset hepatocytes and induced pluripotent stem cell-derived hepatocyte-like cells support HBV and more efficient woolly monkey HBV (WMHBV) infection. Adapted chimeric HBV genome harboring residues 1–48 of WMHBV preS1 generated here led to a more efficient infection than wild-type HBV in primary and stem cell derived marmoset hepatocytes. Collectively, our data demonstrate that minimal targeted simianization of HBV can break the species barrier in small NHPs, paving the path for an HBV primate model.

Hepatitis B virus (HBV), the prototype virus of the *Hepadnaviridae* family, is a hepatotropic virus that chronically infects over 257 million individuals[1,2]. Chronic HBV infection is a major cause of liver fibrosis, cirrhosis, and hepatocellular carcinoma (HCC)[3]. The currently available antiviral options of reverse transcriptase inhibitors and type I interferon (IFN) therapy can suppress viremia but rarely achieve a cure[4].

HBV's highly restricted host tropism has been a long-standing bottleneck for mechanistic dissection of HBV pathogenesis and the systematic assessment of novel therapeutic approaches. Chimpanzees are the only species besides humans naturally susceptible to HBV, but the NIH moratorium on their use in federally funded research[5] has

created an urgent need for new animal models. In the quest for an animal model for HBV infection, different host adaptation approaches have been taken in addition to searching for surrogates based on hepadnaviruses genetically similar to human HBV (reviewed in[6]). Current rodent models suffer from considerable drawbacks. HBV transgenic (tg) mice do not faithfully recapitulate the inflammatory environment caused by bona fide viral infection while human sodium taurocholate co-transporting peptide (hNTCP)—the HBV receptor—tg mice support HBV uptake but not viral replication[7]. Human liver chimeric mice are susceptible to several hepatotropic pathogens, including HBV[8], but their highly immunocompromised status limits

[1]Department of Molecular Biology, Princeton University, Princeton, NJ 08544, USA. [2]Research Computing, Office of Information Technology, Princeton University, Princeton, NJ 08544, USA. [3]Memorial Sloan-Kettering Cancer Center, 1275 York Avenue, New York, NY 10065, USA. [4]Visikol, Inc., Hampton, NJ 08827, USA. [5]Division of Gastroenterology and Hepatology, Department of Medicine, Weill Cornell Medicine, New York, NY 10065, USA. [6]National Emerging Infectious Diseases Laboratories, Boston University, Boston, MA 02118, USA. [7]Department of Pathology and Laboratory Medicine, Boston University Chobanian & Avedisian School of Medicine, Boston, MA 02118, USA. ✉e-mail: aploss@princeton.edu

their utility. The latter can be addressed in part by co-engraftment with components of a human immune system (HIS) in a single xenorecipient[9–12], but the composition and functionality of the engrafted HIS do not yet fully approximate that of humans[13]. Surrogate models such as ducks and woodchucks have been extensively used in HBV research, but the hepadnaviruses causing infections in these species, although similar, are not identical to HBV. Infection of smaller NHPs would be an attractive alternative given their genetic similarity to humans. Although one study suggested that geographically isolated Mauritian cynomolgus macaques may support HBV replication[14], these data have not been independently reproduced.

The apparent block in interspecies transmission can be largely attributed to differences in the amino acid (AA) sequence of the HBV receptor hNTCP. In fact, it was shown that AAs 84–87 and 157–165 of hNTCP contain residues critical for HBV entry, and fine-mapping experiments pinpointed AA 158 in particular[15,16]. However, the contributions of other residues in these regions to HBV uptake remain incompletely understood. To overcome this interspecies block in HBV entry, hNTCP has been expressed in mice[7,17,18] and non-human primates (NHPs)[19]. Although one or more post-entry blocks prevent the completion of the HBV life cycle in hNTCP tg mice[7], adenovirus (Ad)-mediated delivery of hNTCP in rhesus macaques (RMs) results in acute, transient infection[19], which can progress to chronicity if the animals are severely immunocompromised[20]. The latter result indicates that overcoming the entry block in small NHP species may suffice to break the species barrier.

Extensive efforts to identify other primate viruses in the same genus as HBV (Orthohepadnavirus)[21] led to the discovery of wooly monkey HBV (WMHBV)[22], for which an infectious clone has been generated[23], and orthohepadnaviruses from chimpanzee[24] and gibbon[25] that both cluster genetically with HBV. Given that wooly monkeys are an endangered species, their experimental infection with WMHBV is not feasible. It was previously demonstrated that squirrel monkeys exhibit prolonged WMHBV viremia lasting 6–8 months[26], suggesting that this New World monkey (NWM) species may be a suitable model for testing HBV therapeutics. However, some Saimiri species (e.g. Saimiri oerstedii) squirrel monkeys have also been placed on the list of endangered species and are not readily accessible.

In this study, we determined key residues of hNTCP responsible for viral binding and internalization. We identified these residues by systematic genetic and structural comparisons between NTCP orthologues from Old World monkeys (OWM), NWM and prosimian species. Based on these results, we reasoned that marmoset hepatocytes would likely be susceptible to hepadnavirus infection. Indeed, HBV, and to an even greater extent WMHBV, could infect marmoset hepatocytes. Replacing the first 48 residues of HBV preS1 with the equivalent sequence of WMHBV resulted in a chimeric virus (HBV/WMHBV preS1[1–48]) that can robustly infect primary marmoset hepatocytes and iPSC-derived hepatocyte-like cells.

## Results

### NTCP polymorphism between humans and NHPs
We reasoned that a systematic analysis of the molecular determinants facilitating HBV preS1 binding, and ultimately, viral entry, would enable us to identify NTCP orthologues from NHP species that naturally support hepadnavirus uptake. Thus, we performed a phylogenetic analysis of NTCP orthologues from primate, tree shrew, bat, rodent and other species (Supplementary Fig. 1). Expectedly, OWM (97%), NWM (93.4%) and prosimian (87.7%) orthologues were genetically most closely related to human NTCP. NTCP of tree shrews—a species susceptible to HBV[27]—was more closely related to that of primates than rodents and other species. NTCP of OWMs, specifically rhesus and cynomolgus macaques, and NWMs, squirrel monkey (SqM) and marmoset, differed from hNTCP primarily in two regions spanning residues 157–165 and 84–87, respectively (Fig. 1A). Residues G158 for

OWM and N87 for mouse NTCP have previously been implicated in governing HBV entry restriction in these species[28]. Consistently, HepG2 cells expressing human or NWM, but not OWM or prosimian, NTCP tagged with red fluorescent protein (RFP) (Fig. 1B and Supplementary Fig. 2A, B) supported binding of myristoylated HBV preS1[2–48]-FITC peptide (Supplementary Fig. 2C, D). This interaction was specific to the HBV envelope as it could be blocked using the HBV entry inhibitor Myrcludex B (MyrB)[29] (Fig. 1C). Fluorescence confocal microscopy analysis confirmed that the HBV peptide and NTCP colocalized on the cell membrane (Supplementary Fig. 2E). Notably, none of the OWM, NWM or prosimian NTCP orthologues supported significant HBV infection (Fig. 1D).

### Mutagenesis of NHP NTCPs identifies key residues for HBV infection
Recently, cryo-EM structures of NTCP have been reported[30–32], which enabled us to gain more mechanistic insights into how polymorphisms in residues 84–87 and 157–165 affect HBV infection. Human NTCP is composed of 9 transmembrane (TM) helices, of which TM 1, 3, 5, and 8 form the functional tunnel (Fig. 2A). Residues 84–87 are part of a loop domain that is responsible for the extracellular interaction between HBV preS1 and NTCP, while the tunnel pocket, which contains AAs 157–165, is the competitive cavity for preS1 docking and bile acid transport (Fig. 2A). We predicted structures of select NTCP orthologues by AlphaFold[33] and the predictions are with high confidence and accuracy (pLDDT>85 except some loop domains in the NTCP) (Supplementary Fig. 3). The predicted NTCP structures for OWM, NWM and the variants harboring humanizing point mutations (OWM[R158G], OWM[R158G, P165L], SqM[Q84R] and SqM[K87N]) adopted conformations similar to the human cryo-EM structures and none of the mutations disrupted the overall architecture of NTCP characterized by the alignment distance (Supplementary Fig. 3). To elucidate the molecular determinants of the restricted HBV host tropism in OWM and NWM, we introduced humanizing mutations stepwise in the 157–165 and 84–87 regions and analyzed their effects on the NTCP structure and its ability to support HBV uptake (Fig. 2B; Supplementary Fig. 4A).

Functionally, R158G enables binding of preS1-derived peptides to OWM-NTCP, whereas G158R abrogates preS1 binding to SqM-NTCP detected by flow cytometry and confocal imaging (Fig. 2C, D; Supplementary Fig. 4B, C), underscoring that G158 is necessary for HBV binding. Interestingly, double humanizing mutations at residues 158 and 165 increased HBV's ability to bind (Supplementary Fig. 5A) and infect HepG2 cells, as evidenced by an over 30-fold increase in HBeAg compared to the R158G single mutant (Fig. 2E). However, since the binding is the prerequisite for infection, the Q84R and N86K mutations in OWM NTCP did not support HBV infection without the R158G mutation (Supplementary Fig. 6). For NWM NTCP, humanizing residue 84, and even more so 87, rendered HepG2 cells susceptible to HBV, and mutating residue 86 had an additive enhancing effect (Fig. 2E), suggesting residues 84–87 are functional for HBV internalization. HBV infection mediated by the gain-of-function NTCP mutations was further confirmed by formation of cccDNA and inhibition of infection with Myrcludex B (Supplementary Fig. 5B, E). Notably, humanizing residues from 84–87 within NWM NTCP led to an increase in HBV infectivity and a reduction in preS1-derived peptide binding (Supplementary Fig. 5A, C and D).

### Structural basis for the differing HBV susceptibility of NTCP orthologues
The structure of hNTCP was divided into core (TM2–4 and TM7–9) and panel (TM1 and TM5&6) domains based on the resolved crystal structure of the prokaryotic homologue apical sodium-dependent bile acid transporter (ASBT)[31,34]. A nine-residue motif (AAs 157–165) in the N-terminal of TM5 in the panel domain restricts OWM-NTCP from mediating HBV infection[35] (Fig. 2A). TM5 together with TM1, TM3 and

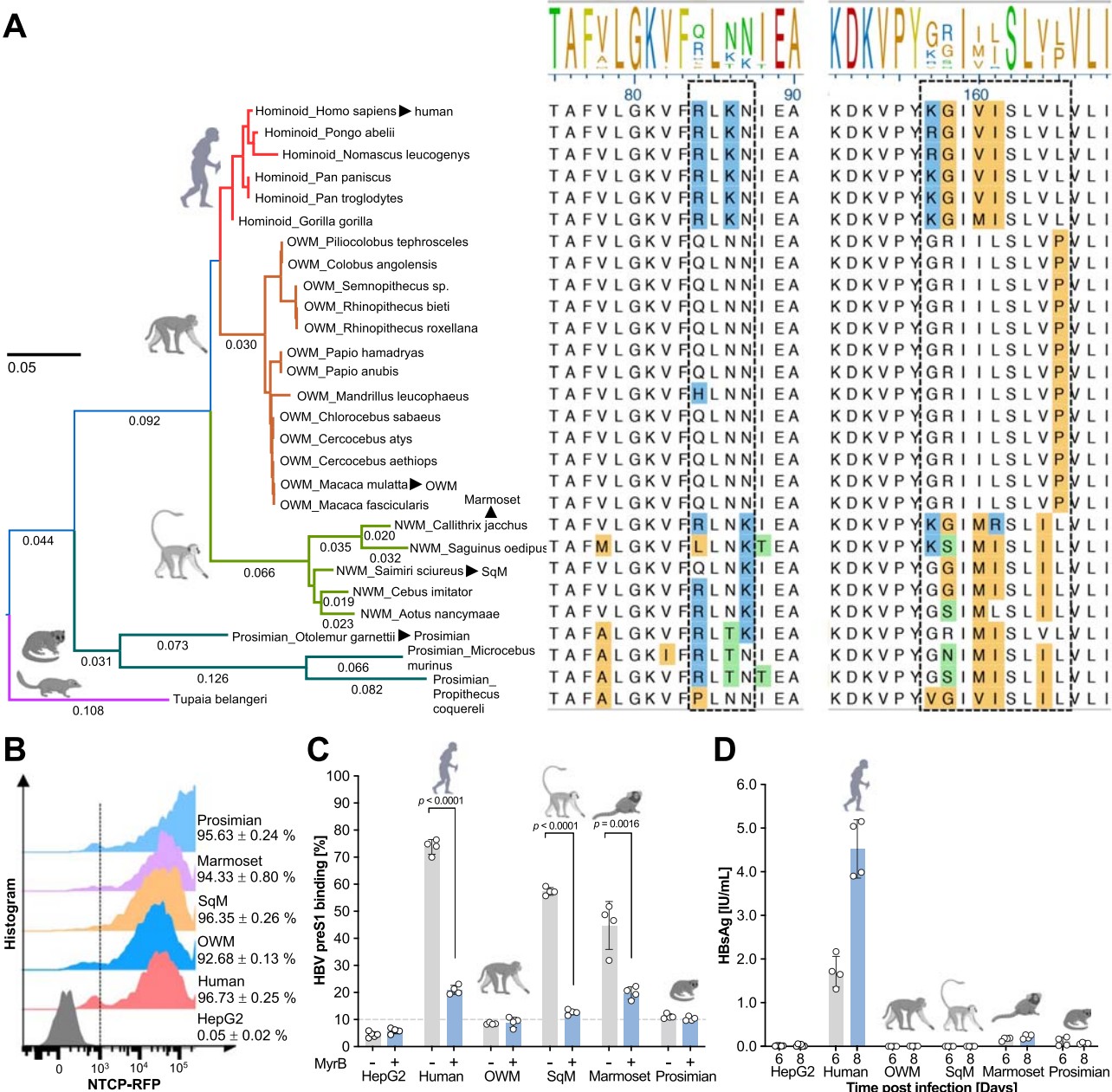

**Fig. 1 | HBV has a narrow species tropism in primates. A** Phylogenetic analysis and sequence alignment of two important regions (highlighted with dash rectangles) of NTCPs in humans, primates, and tree shew. Arrows indicate the NTCP orthologs used in this study. Differential residues were highlighted by color through MegAlign Pro (DNASTAR Inc., Madison, WI) during the phylogenetic assay. **B** Ectopic NTCP (tagged with RFP) overexpression of different species, respectively. The mean and std. error of mean are based on 4 biologically independent experiments. The dash line indicates the gating for background fluorescence intensity. **C** HBV preS1 binding assay on HepG2 cells expressing NTCP from human, OWM, squirrel monkey, marmoset and prosimian species. Quantification of 5′ myristoylated and 3′ FITC-labeled HBV preS1 AAs 2–48 peptide binding (200 nM) in the presence or absence of Myrcludex B (750 nM) by flow cytometry. Bars depict the mean of 4 biologically independent experiments and error bars represent s.e.m. The dash line indicates the defined baseline of the assay based on the control groups. Unpaired *t* test, two-tailed. **D** HBsAg detection in supernatants of HBV-infected (MOI = 8000) HepG2 cells expressing NTCP from human, OWM, squirrel monkey, marmoset or prosimian species. Bars depict mean of 4 biologically independent experiments (*n* = 6 for control group) and error bars represent s.e.m. Primate icons were created with BioRender.com. OWM old world monkey, SqM squirrel monkey. Source data are provided as a Source Data file.

---

TM8 forms the functional tunnel of NTCP (Fig. 3A). The tunnel cavity is responsible for HBV binding and bile salt uptake via transitions between inward-facing and open-pore states[31,32]. Conformationally, R158 in OWM-NTCP introduces a bulky side chain in the functional tunnel, presumably obstructing the cavity and blocking HBV binding (Fig.3A). P165 in OWM-NTCP vs L165 in hNTCP may disrupt the formation of a helix that stabilizes a loop structure between TM4 and TM5 (Fig. 3B) and helps control the distance and conformation between the core and panel domains to maintain the function of the tunnel and facilitate HBV infection. Humanizing another patch in the TM2-3 loop allows NWM-NTCP to adopt structures with physicochemical properties more similar to hNTCP at the side chains (Fig. 3C). Overlaying the NTCP orthologue structures on hNTCP revealed that R158G and P165L double mutations in OWM-NTCP showed the highest structural similarity with hNTCP as characterized by the root mean squared deviation (RMSD) (Fig. 3D; Supplementary Fig. 3). Notably, among wild type (WT) NTCPs from OWM and NWM, marmoset structure had the lowest RMSD from hNTCP (Fig. 3D), suggesting marmoset hepatocytes may

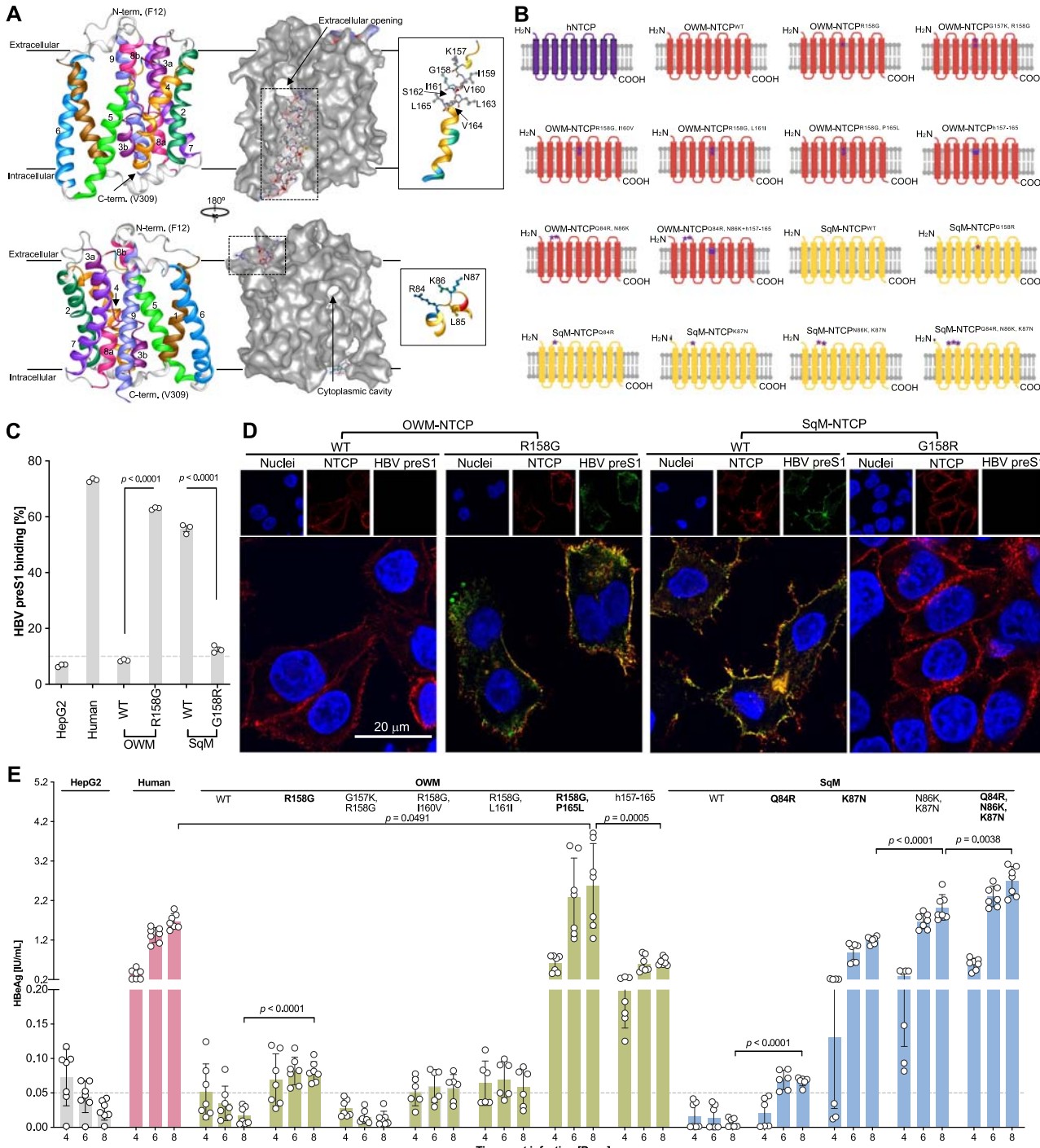

**Fig. 2 | NTCP polymorphisms in NHPs determines HBV susceptibility. A** Ribbon diagrams (left) and molecular surface (right) representation of hNTCP structure showing forward and back views based on released cryo-EM data (PDB: 7PQQ). Residues related to HBV preS1 binding and the NTCP functional cavity are labeled. **B** Schematics (Created with BioRender.com.) for hNTCP, OWM NTCP, NWM NTCP and their humanized mutants. **C** HBV binding assay on HepG2 cells expressing NTCP orthologues with or without mutations by flow cytometry using 5′ myr-istoylated and 3′ FITC-labeled HBV preS1 AAs 2–48 peptide (200 nM) in the presence or absence of Myrcludex B (750 nM). The dashed line indicates the defined baseline of the assay based on the control groups. Gray bars depict mean of 3

biologically independent experiments and error bars represent s.e.m. Unpaired *t* test, two-tailed. **D** HBV binding assay on HepG2 cells expressing NTCP orthologues with or without mutations by fluorescent microscopy imaging using 5′ myr-istoylated and 3′ FITC-labeled HBV preS1 AAs 2–48 peptide (200 nM). **E** HBeAg detection in supernatants of HBV-infected (MOI = 6000) HepG2 cells expressing different NTCP orthologues. Bars depict mean of 7 independent experiments (*n* = 6 for OWM-NTCP$^{R158G, I160V}$, OWM-NTCP$^{R158G, L161I}$ and SqM-NTCP$^{Q84R}$ groups) and error bars represent s.e.m. The dashed line indicates the low limit of the assay based on the control groups. OWM old world monkey, SqM squirrel monkey, WT wild type. Unpaired *t* test, two-tailed. Source data are provided as a Source Data file.

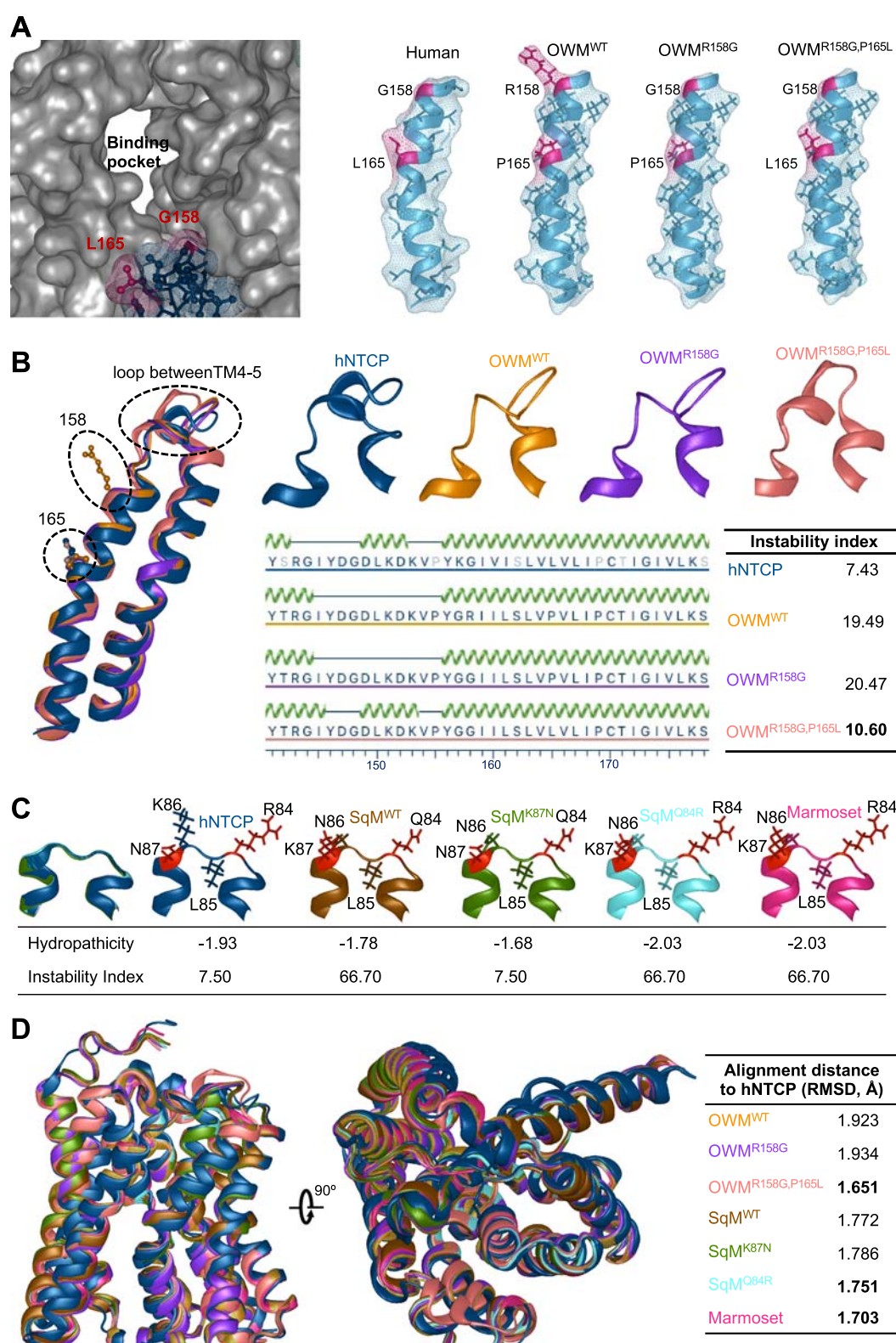

**Fig. 3 | Structural consequences of humanizing residues in OWM and NWM NTCP. A** Molecular surface model showing the hNTCP functional tunnel (PDB: 7PQQ) and the TM5 region in different NTCP orthologues. **B** Structure alignment of the TM4-5 region of hNTCP (PDB: 7PQG), OWM^WT, OWM^R158G and OWM^R158G, P165L. A helix in the TM4-5 loop in hNTCP and OWM^R158G, P165L can increase NTCP stability. **C** NTCP structures with side chains in residue 84–87 for hNTCP (PDB: 7PQG),

marmoset, squirrel monkey (SqM) and their variants. The hydropathicity and instability index indicating the biophysical properties of residues 84−87 were calculated in Protein 3D (DNASTAR Inc., Madison, WI). **D** Structure alignment of different NTCPs (hNTCP structure from PDB: 7PQG). Similarity assay conducted by root mean squared deviation (RMSD). OWM old world monkey, SqM squirrel monkey, WT wild type, Cyno cynomolgus machaca, TM transmembrane.

support hepadnavirus infection. Collectively, these data indicate which NTCP polymorphisms may contribute to the species barrier in OWMs and NWMs and provide structural insights into HBV susceptibility.

### Marmoset hepatocytes support the full WMHBV and post-entry HBV life-cycles

We next sought to identify a NWM species that could naturally support HBV infection. We previously demonstrated that primary SqM hepatocytes are susceptible to WMHBV in vivo[26], but since SqM species are endangered they are not suitable as an animal model. Notably, we found marmoset NTCP supports HBV binding (Fig. 1B), WMHBV preS1 binding (Fig. 4A, B and Supplementary Fig. 7A) and WMHBV infection (Fig. 4C). Furthermore, among the NHP NTCP orthologues examined, WT marmoset NTCP was most similar to hNTCP, with a RMSD of 1.703 Å (Fig. 3D). Thus, to analyze the extent to which primary marmoset hepatocytes (PMH) support the hepadnaviral life cycle, we established self-assembling co-cultures (SACCs)[36] of PMH and murine non-parenchymal stromal cells (Fig. 4D), which can maintain the highly differentiated phenotype of primary hepatocytes for at least 4 weeks (Supplementary Fig. 7B). Lentiviral delivery of a 1.3x HBV genome into SACC-PMHs resulted in HBV replication (Fig. 4E). Ectopic expression of hNTCP followed by HBV inoculation resulted in robust infection (Fig. 4F). Consistent with our preS1 binding data (Fig. 4A), WMHBV can robustly infect PMHs (Fig. 4G).

### Generation of simian-tropic HBV by targeted viral adaptation

Since PMHs can support HBV replication and the entire WMHBV life cycle, we next attempted to derive an HBV-based virus that included only small portions of WMHBV. We replaced residues 1–48 of HBV preS1, the key regions mediating HBV entry[37,38], with the equivalent WMHBV sequence that differs by 16 AAs (Supplementary Fig. 7C), yielding a genome termed HBV/WMHBV preS1[1–48] (Fig. 5A). Of note, replacement of AAs 1–48 did not impact the other overlapping open reading frames (ORFs) of the HBV genome. Transfection of HBV/WMHBV preS1[1–48] into HepG2 cells resulted in HBV DNA copy numbers and HBsAg levels in the supernatants equivalent to those of the parental HBV and WMHBV genomes, while WMHBV yielded higher HBeAg levels ($p < 0.0001$) (Supplementary Fig. 7D). All three viruses had similar biophysical properties, as evidenced by HBV DNA and HBsAg being detectable in the same fraction of a sucrose gradient (Fig. 5B; Supplementary Fig. 7E). Negative-stain electron microscopy revealed structures consistent with Dane particles (Fig. 5C). Collectively, these data demonstrated that our chimeric HBV/WMHBV preS1[1–48] is replication-competent and can form mature viral particles.

Next, we evaluated binding and internalization of the HBV, WMHBV, and chimeric virus in HepG2 cells expressing marmoset NTCP (Fig. 5D). All three viruses could bind marmoset NTCP (Fig. 5E), but the HBV/WMHBV preS1[1–48] chimera was more readily capable of entering cells than WT HBV ($p < 0.0314$) (Fig. 5F). Thus, we investigated whether HepG2 cells expressing marmoset NTCP could support infection with the viruses. We confirmed by immunofluorescence (IF) that the HBV core protein (HBc) antibody could also be used to detect the core protein of WMHBV and the chimeric virus following transfection of the infectious clones (Supplementary Fig. 8). Then, by using WMHBV as a positive control for infection on marmoset NTCP expressing HepG2 cells, we determined that HepG2 cells expressing marmoset NTCP could support some level of HBV infection, but HBV/WMHBV preS1[1–48] resulted in more HBcAg+ cells (Supplementary Fig. 9). HBsAg levels were subsequently detected in the supernatants after 6 and 8 days of infection (Fig. 5G). We also found detectable HBV cccDNA inside of the cells (Fig. 5H) using a Hirt DNA extraction, T5 exonuclease digestion and HBV cccDNA specific qPCR determination as previously described[39,40]. We confirmed the presence of HBV DNA

replication intermediates by Southern blotting. Our data are consistent with the HBsAg and cccDNA qPCR data confirming higher levels of HBV DNA in HBV/WMHBV preS1[1–48] group (Fig. 5I). To provide additional evidence that marmoset NTCP expressing cells were infected, we employed a multiplexed HBV RNA in situ hybridization (ISH), HBc and NTCP immunohistochemistry (IHC) assays (Supplementary Fig. 10). HBV RNA and HBcAg were readily detectable in HepG2-hNTCP cells infected with HBV, thus validating the assay. HBV RNA and HBc were also detected in marmoset NTCP-expressing cells infected with HBV or HBV/WMHBV preS1[1–48], respectively (Fig. 5J).

### Simian-tropic HBV establishes more robust infections in PMHs

Consistent with our previous results (Fig. 3G), WMHBV robustly infected SACC-PMHs, and inoculation of SACC-PMHs with HBV/WMHBV preS1[1–48] or HBV resulted in consistent HBeAg production in the culture supernatants (Fig. 6A). The infectivity of the chimeric virus was significantly higher than that of HBV, as evidenced by higher intracellular HBV DNA ($p < 0.0275$), pre-genomic RNA (pgRNA; $p < 0.0038$) (Fig. 6A), and covalently closed circular DNA (cccDNA; -100-fold) (Supplementary Fig. 7F). To further substantiate our findings, we generated cultures of marmoset induced pluripotent stem cell (iPSC)-derived hepatocyte-like cells (iPS-MHLCs) (Fig. 6B) using a directed differentiation approach[41]. The stepwise differentiation of the iPSCs (Supplementary Fig. 11), and hepatic differentiation and specification of the iPS-MHLCs were characterized by hepatocyte-specific genes expression (Fig. 6C) and liver-specific functions including urea secretion and cytochrome P450 (CYP) activities (Fig. 6D). Inoculation of iPS-MHLCs with HBV or HBV/WMHBV preS1[1–48] resulted in successful infection and HBeAg production (Fig. 6E, left). The adapted HBV/WMHBV preS1[1–48] chimera established more robust infections than HBV as evidenced by significantly higher levels of HBeAg ($p = 0.0029$) in the culture supernatants, and HBV DNA ($p = 0.0027$) and pgRNA ($p = 0.0179$) copies in cell lysates (Fig. 6E).

## Discussion

This study represents a major step forward in the development of a small animal model for HBV infection. Combining comprehensive genetic mutational analysis with recently emerging structural information and molecular modeling laid the foundation for our targeted viral engineering approach. Previous mutational analysis identified key residues between NTCP orthologues that support HBV entry (human) vs those that do not (e.g. rhesus macaque and mouse)[16,28,35,42]. Humanizing NTCP residues 84–87 in mouse cell lines[43] and mice[17] facilitates HBV glycoprotein-mediated uptake. Results from our analysis demonstrate that several of the residues in this region also block HBV infection in NWMs. Although humanizing Q84 resulted in a small but reproducible enhancement of HBV infection, additional humanization at N86 and K87 resulted in a 10–20 fold increase. Notably, we found that humanizing residues Q84R, K87N, N86K/K87N, and Q84R/N86K/K87N in SqM NTCP facilitated HBV infection but decreased HBV binding. Based on the structural data, the AA84–87 motif is part of a loop domain between transmembrane 2-3 and bile acids do not compete with HBV at this region[30]. Our results suggest that the AA84–87 motif is mainly responsible for HBV entry. Conceivably, interactions between the HBV envelope and NTCP within this region causes topological/conformational changes that ultimately trigger endocytosis.

Residue 158 within TM5 of NTCP is exposed to the channel responsible for bile acid transport and differs between rhesus macaques and humans. Changing the arginine in rhesus NTCP to glycine facilitates binding and uptake[42]. Our data confirmed these prior observations, but our extensive mutational analysis demonstrated that the additional P165L mutation further enhanced HBV infection by 10–15 fold. Notably, mutating R158G and P165L led to a greater increase in HBV infection than replacing the entire 157–165 sequence with the human counterpart, suggesting that other residues may

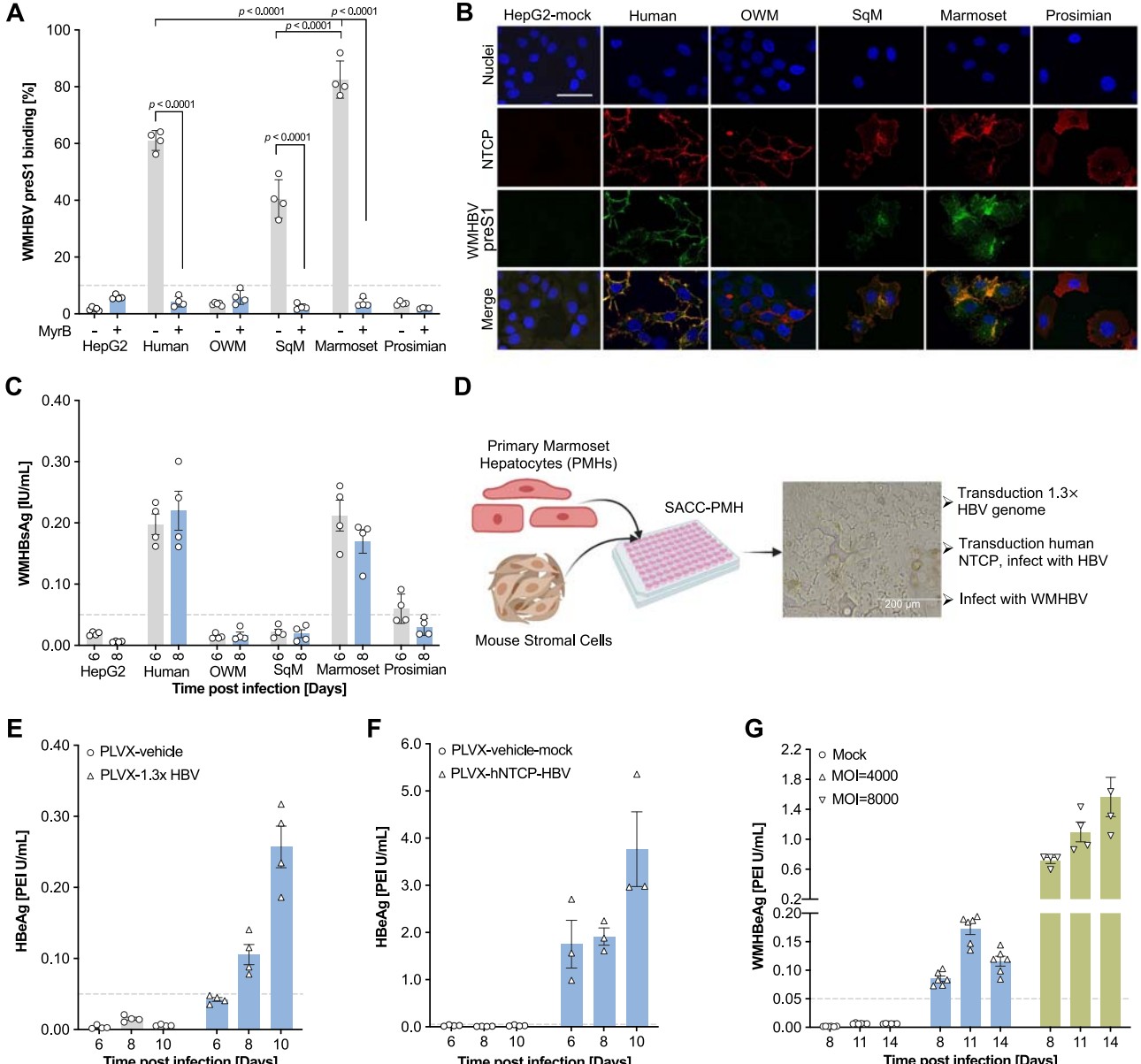

**Fig. 4 | WMHBV can infect HepG2 cells expressing marmoset NTCP and primary marmoset hepatocytes. A** WMHBV binding assay on HepG2 cells expressing NTCP from human, OWM, squirrel monkey, marmoset and prosimians. Quantification of 5′ myristoylated and 3′ FITC-labeled HBV preS1 AAs 2–48 peptide binding (200 nM) in the presence or absence of Myrcludex B (750 nM) by flow cytometry. The dash line indicates the defined baseline of the assay based on the control groups. Bars depict mean of 4 biologically independent experiments and error bars represent s.e.m. Unpaired $t$ test, two-tailed. **B** Confocal imaging showing the binding of WMHBV preS1 (200 nM) to different NTCPs. The experiments were repeated twice independently with similar results and the representative images were shown. The scale bar is 20 μm. **C** WMHBsAg detection in supernatants of WMHBV-infected (MOI = 8000) HepG2 cells. Bars depict mean of 4 biologically independent

experiments and error bars represent s.e.m. The dash line indicates the low limit of the assay based on the control groups. **D** Schematic (Created with BioRender.com.) of self-assembling primary marmoset hepatocyte (SACC-PMH) co-cultures system for HBV/WMHBV replication or infection. Quantification of HBeAg in supernatants of primary marmoset hepatocytes (PMHs) transduced with a 1.3x HBV genome (**E**); transduced with hNTCP and infected with HBV (**F**); or infected with WMHBV (**G**). Bars depict mean of biologically independent experiments ($n = 4$ in **E**; $n = 4$ in PLVX-vehicle-mock group and $n = 3$ in PLVX-hNTCP-HBV group in **F**; $n = 6$ in mock and MOI = 4000 groups and $n = 4$ in MOI = 8000 group in **G**) and error bars represent s.e.m. The dashed lines indicate the low limit of the assay based on the control groups. Source data are provided as a Source Data file.

actually have an inhibitory effect. While the R158G mutation introduces a smaller side chain that presumably enables preS1 binding, the P165L mutation may contribute to stabilizing the loop structure between TM4 and TM5 by forming a helix. As a result, the core and panel domains are conceivably brought into a conformation that maintains the functional tunnel and facilitates HBV infection. To gain further insights into the function of residue 158 in the HBV entry process, we introduced mutations that enabled or abrogated HBV binding—R158G in OWM and G158R in NWM NTCP, respectively. These

results revealed that HBV can bind to NWM and OWM NTCP regardless of the residues in the 84–87 region.

Notably, we found HBV could infect PMHs and the iPS-derived marmoset HLCs. In these cells, simianized HBV/WMHBV preS1[1–48] chimeric virus showed higher infectivity, but the HBeAg levels in the supernatant were lower than those observed after infection with WMHBV. On the other hand, we did not see significant increases in HBeAg over time in the supernatants after infection, even though we detected HBV RNA- and HBc-positive cells by ISH, IHC and IF, as well as

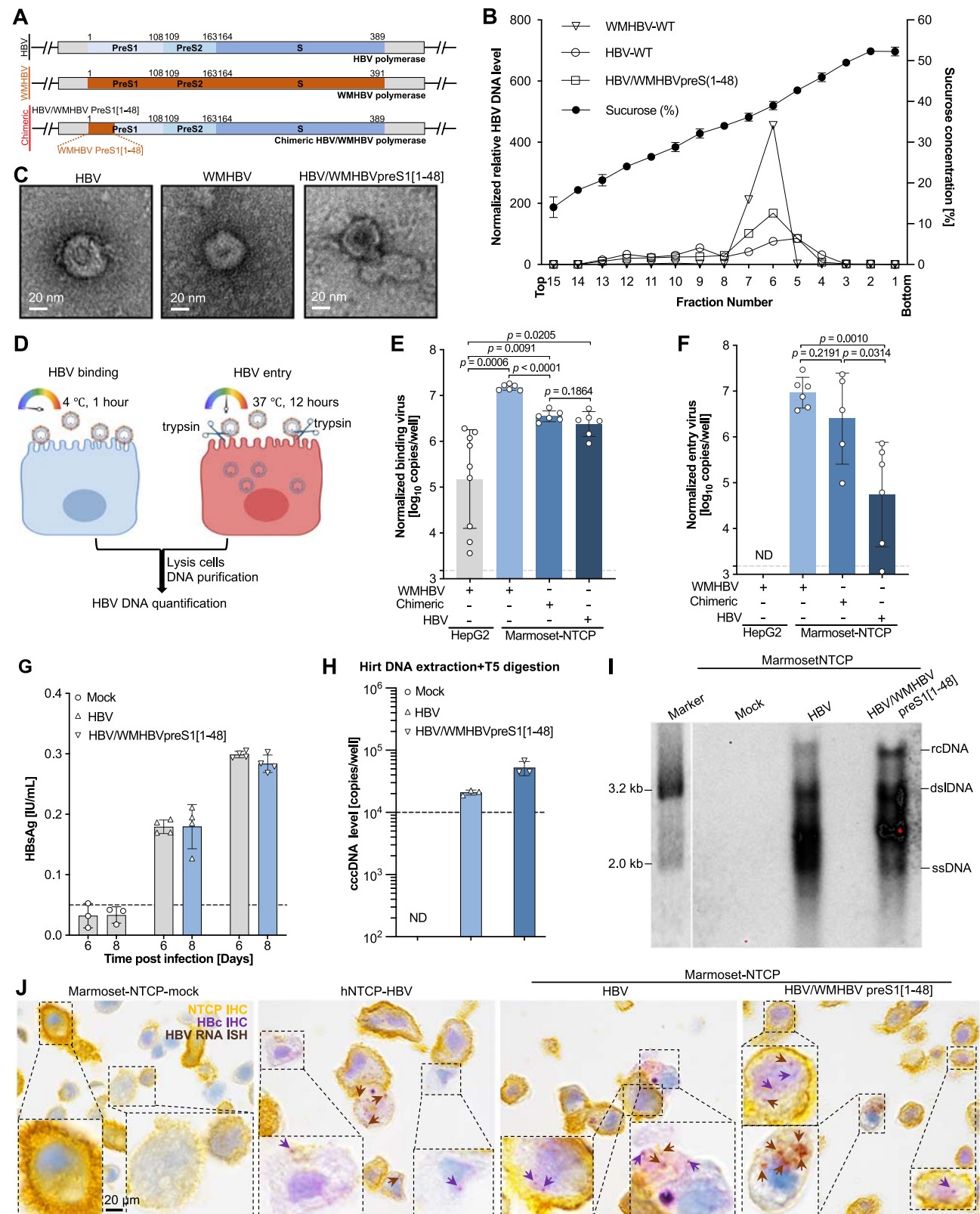

significant HBsAg levels in the supernatant for HBV and HBV/WMHBV preS1[1–48] virus infection of HepG2 cells expressing marmoset NTCP. This could be similar to the effect seen in primary human hepatocytes vs NTCP-reconstituted human hepatoma cells. In the latter, HBV tends to have lower infectivity due to the dysregulated growth[44] and the instability of NTCP expression and localization in hepatoma cells[45]. Meanwhile, WMHBV per se can secrete higher levels (~3.8-fold) of

HBeAg in the supernatant than HBV or HBV/WMHBV preS1[1–48] (Supplementary Fig. 7D). While analyzing simultaneously NTCP and HBc or HBV RNA in the HBV infection experiments, we noticed that NTCP expression was lower in HBc and HBV RNA positive cells. Putatively, one possibility for lower NTCP expression on the membrane is that HBV is believed to enter the cells by NTCP-mediated endocytosis[46]. Following internalization of NTCP, receptors may be

**Fig. 5 | Generation and characterization of chimeric WMHBV/HBV viruses.**
**A** Schematic of HBV, WMHBV and HBV/WMHBV preS1[1–48] chimeric virus genomes. The preS1 open reading frame together with preS2 and S inside of the polymerase gene are shown. **B** Biochemical characterization of WMHBV, HBV and HBV/WMHBVpreS1 (1–48) chimeric viruses by sucrose density gradient centrifugation. HBV DNA levels were shown as mean of two biologically independent experiments. Sucrose concentration in each fraction was tested three times and was shown as mean ± s.e.m. **C** Transmission electron microscopic images of HBV, WMHBV and HBV/WMHBV preS1[1–48] chimeric virions. A 2 µl sample from fraction 6 separated by sucrose density gradient centrifugation was used. The experiments were repeated twice independently with similar results and the representative images were shown. **D** Schematic (Created with BioRender.com.) representation of the virion binding and entry assay. HBV, WMHBV and HBV/WMHBV preS1[1–48] chimeric virion binding (**E**) and entry (**F**) using HepG2 cells expressing marmoset NTCP. HepG2 cells lacking NTCP expression were used as controls. Bars depict mean of 6 biologically independent experiments (*n* = 9 for control group in (**E**) and *n* = 5 in chimeric virus group in (**F**) and error bars represent

s.e.m. Unpaired *t* test, two-tailed. The dash lines indicate the low limit of the detection. ND not detected. **G** HBsAg detection in supernatants of HBV- or HBV/ WMHBV preS1[1–48]-infected (MOI = 8000) HepG2 marmoset NTCP cells. Bars depict mean of 4 biologically independent experiments and error bars represent s.e.m. The dashed line indicates the low limit of the detection. **H** HBV cccDNA detection by Hirt DNA extraction, T5 exonuclease and HBV cccDNA specific qPCR. Bars depict mean of 3 biologically independent experiments and error bars represent s.e.m. The dash line indicates the low limit of the detection. ND not detected. **I** Southern bot assay of intracellular HBV DNA replication intermediates after infection. The experiments were repeated twice independently and the uncropped gels were shown in the supplementary Fig. 12. **J** Multiplexed HBV RNA in situ hybridization (ISH), HBc and NTCP immunohistochemistry (IHC) assays using formalin-fixed, paraffin-embedded (FFPE) cell pellets. Representative positive signals were indicated by arrows (HBV RNA, brown; HBc, purple). At least 3 images from different fields were taken from which the representative images are shown. Source data are provided as a Source Data file.

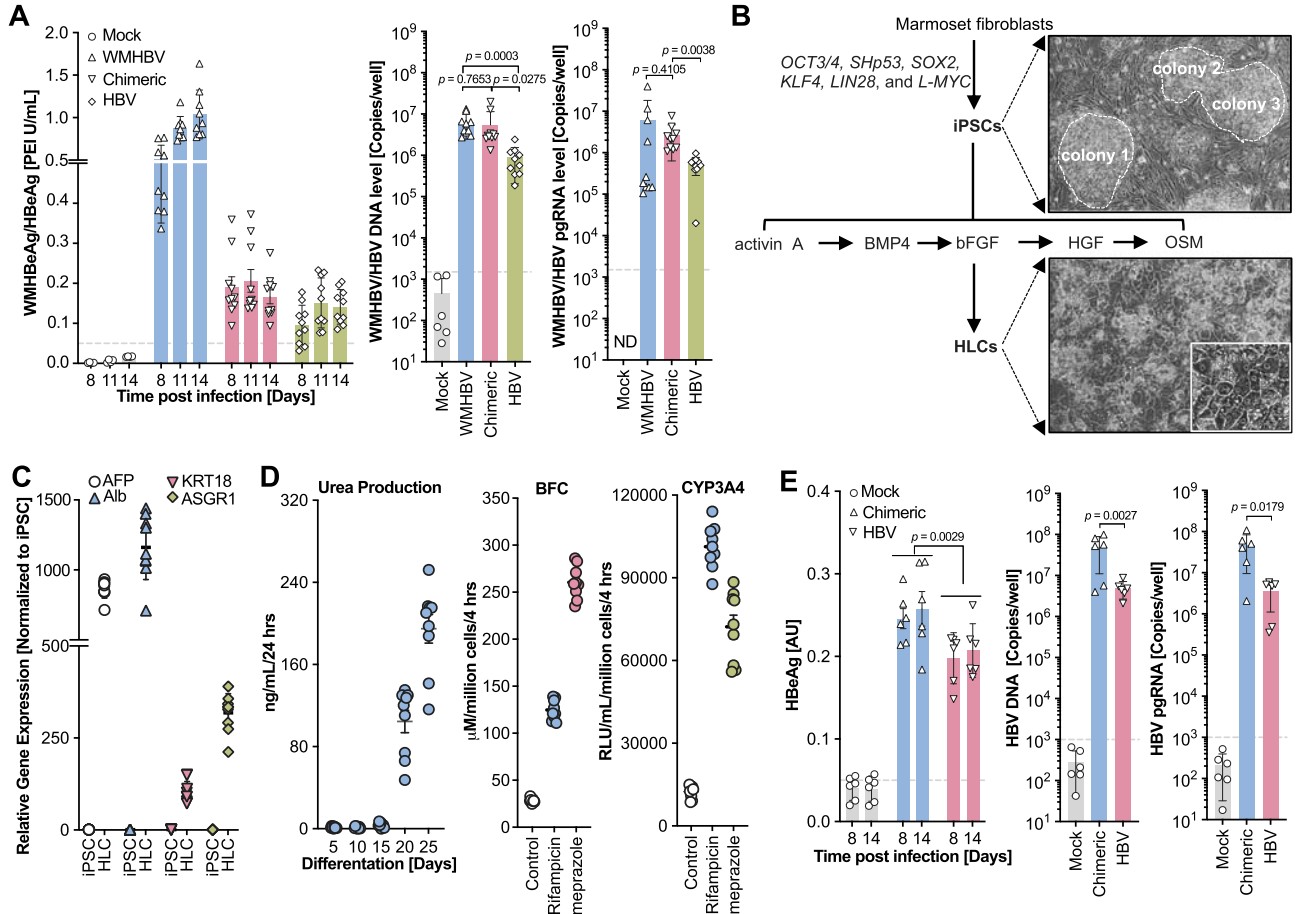

**Fig. 6 | Simian-adapted HBV causes more robust infections in primary marmoset hepatocytes and induced hepatocyte-like-cells.** **A** Hepadnavirus infections of SACC-PMH cultures. WMHBeAg/HBeAg levels (left) were detected in the supernatants at days 8, 11, and 14 post-infection. SACC-PMHs that had not been exposed to any virus were used as controls. Intracellular WMHBV/HBV DNA (middle) and WMHBV/HBV pgRNA (right) were detected by qPCR and RT-qPCR, respectively, at the endpoints. The dash lines indicate the low limit of the detection. Bars depict mean of 10 biologically independent experiments (for mock group, *n* = 4 for HBeAg detection and *n* = 6 for DNA/pgRNA detection) and error bars represent s.e.m. Two tailed unpaired *t* test. **B** Schematic representation of iPS-derived marmoset hepatocyte-like cells (iPS-MHLC). The corresponding representative images at different stages for iPS cells (iPSCs) and HLCs are shown,

respectively. **C** Hepatic gene expression levels in the HLCs compared with those in the undifferentiated iPSCs. Data are presented as the mean of 9 biologically independent experiments and error bars represent s.e.m. **D** Functional assessment of the iPS-MHLCs. Data are presented as the mean of 9 biologically independent experiments and error bars represent s.e.m. **E** HBV and HBV/WMHBV preS1[1–48] chimeric virus infections of iPS-MHLCs. At the indicated timepoints, HBeAg was detected in the supernatants (left) by ELISA, and intracellular HBV DNA (middle) and HBV pgRNA (right) by qPCR and RT-qPCR, respectively. Bars depict mean of 6 biologically independent experiments and error bars represent s.e.m. Two tailed unpaired *t* test or Nested *t* test and one-way ANOVA were used for the statistics. The dashed lines indicate the low limit of the detection. Source data are provided as a Source Data file.

retained intracellularly and/or degraded, which would prevent recycling. Another possibility could be the superinfection exclusion: the ability of an established virus infection to interfere with a secondary virus infection. This happens in variety of viruses such as Newcastle disease virus (NDV)[47], Rous sarcoma virus (RSV)[48], bovine viral diarrhea virus (BVDV)[49] and hepatitis C virus (HCV)[50], but the exact underlying mechanisms still need further studies.

Importantly, in our analysis, the marmoset sequence stood out among the NTCP orthologues from various OWM and NWM species as it showed the greatest structural similarity to hNTCP. In line with this observation, marmoset NTCP supported binding and entry of HBV and WMHBV. Our data further demonstrated that swapping the first 48 amino acids of the preS1 region of HBV with those of WMHBV considerably enhanced the ability of this recombinant virus to infect PMHs and HLCs. Future efforts will focus on determining whether introducing similar mutations into infectious clones of genetically diverse HBV genotypes yields chimeric genomes inherently more capable of infecting PMHs. Conceivably, infections with simian-tropic HBV could be further enhanced by selecting for replication-enhancing mutations. HBV has a remarkable genetic plasticity fueled by an error-prone reverse polymerase[51], which results in tremendous genetic diversity, readily acquired viral resistance, escape from immune pressure, and putative mutations that may boost HBV infection in marmoset cells.

Our targeted viral adaptation approach to broaden the host range of HBV complements efforts to adapt host species to create an environment more conducive to HBV infection. While mice would arguably be the most attractive species for a small animal model of HBV infection, there are several blocks in the viral replication cycle upstream of the intranuclear replication stages, including but not limited to HBV entry, capsid disassembly, nuclear import, and deproteination of the relaxed-circular DNA genome[52]. In NHPs and pigs, the block appears to be primarily limited to viral entry[53]. Previous work has shown that vector-mediated expression of hNTCP in rhesus macaques and subsequent inoculation with HBV can result in acute infection[19] that may progress to chronicity when animals are severely immunosuppressed[20]. It remains to be seen whether similar immunosuppressive regimens would have to be applied in marmosets to establish an infection.

Marmosets are a commonly used small NHP model in biomedical research and can be bred more quickly than macaques, as they reach sexual maturity faster and most often give birth to twins[54]. The latter is particularly attractive for genetically controlled studies analyzing the underlying basis of age-dependent susceptibility to HBV infection. Our work thus represents an excellent starting point for establishing marmosets as a potential animal model for HBV infection.

## Methods

### Cell lines
293T (American Tissue Culture Collection, ATCC® Number: CRL-3216TM, Manassas, VA) and HepG2 cell lines (American Tissue Culture Collection, ATCC® Number: HB-8065™, Manassas, VA) were maintained in Dulbecco's modified Eagle medium (DMEM; ThermoFisher, Waltham, MA) base medium supplemented with 10% (vol/vol) fetal bovine serum (FBS) (Sigma Aldrich, St Louis, MO). 293T cells were grown on tissue culture-treated plastic ware (Corning Inc., Corning, NY) and HepG2 cells on type IV collagen-coated plates (Sigma Aldrich, St Louis, MO).

### Generation of SACC-PMHs
Cryopreserved marmoset hepatocytes were obtained from Bioreclamation IVT Inc. (Westbury, NY). The co-culture model consists of a mixture of marmoset hepatocytes and non-parenchymal mouse embryonic fibroblast 3T3-J2 cells (CCL-92, ATCC, Manassas, VA) were built as previously described[36,55]. Briefly, cryopreserved hepatocytes were thawed and centrifuged at 150 × g for 10 min and then the cells were re-suspended in Hμrel plating medium™ (Visikol Inc., Hampton,

NJ). Hepatocyte number and cell viability were assessed using trypan blue exclusion. 3T3-J2 cells were cultured in normal DMEM medium (10% FBS, 200 U/mL penicillin/streptomycin) at 37 °C with 5% CO$_2$. Hepatocytes were seeded at a density of 30,000 cells in each well of a 96-well plate, respectively. 3T3-J2 cells were added the next day at 15,000 per well of 96-well plate, respectively. HμREL™–96 SACC-PMH are distributed by the Visikol, Inc. (Hampton, NJ). Cells were maintained in 100 μl Hμrel maintenance medium™ (Visikol, Inc., Hampton, NJ). Medium was replaced every 2 days. The cells were co-cultured at 37 °C in a 5% CO$_2$ for 10 days prior to HBV infections.

### Construction of NTCP orthologues and their humanized mutants
All NTCP orthologues used in this study were inserted into a consistent pLVX lentiviral backbone upstream of a VSKGE protein linker and the tagRFP reporter. First, for human NTCP (hNTCP, NCBI accession number: JQ814895.1), in the first round of PCR: Round 1, Reaction 1: hNTCP was amplified from pCMV-Sport6 Slc10A1 hNTCP (OpenBiosystems, now Dharmacon, Lafayette CO). A forward primer introducing flanking 5′ XhoI site, 5′ Kozak sequence and 5′ FLAG to the hNTCP, as well as a 3′ linker sequence (GTGAGCAAGGGCGAG) and overlap fragment from the tagRFP coding sequence was used. Round 1, Reaction 2: tagRFP was PCR amplified from custom-made gene block (IDT, San Jose, CA) (a previously functionally characterized version of tagRFP) to introduce a 5′ overhang of overlapping end of the hNTCP ORF and linker sequence, and 3′ MluI site. In the second round of overlap PCR, the two products from the first round of PCR (hNTCP and modified tagRFP) were used as a template with the external primers to yield the final PCR product: XhoI-KOZAK-FLAG-hNTCP-VSKGE-tagRFP-MluI. This PCR product was then infusion cloned (Clontech, Mountain View CA) into a lentiviral backbone vector pLVX which had been digested with XhoI/MluI following the manufacturer's instructions. For CynoNTCP (NCBI accession number: AK240620.1), SqMNTCP (NCBI accession number: XM_003924480.3), marmoset NTCP (NCBI accession number: XM_035260831.1) and prosimian NTCP (*Propithecus coquereli*, NCBI accession number: XM_012656155.1), corresponding ORFs were amplified from custom made gene blocks (IDT, San Jose, CA) by using forward primer with flanking 5′ XhoI site, 5′ Kozak sequence and 5′ FLAG and reverse primer with flanking 3′ BsrGI site, as well as gene specific sequences to CynoNTCP, SqMNTCP, marmoset NTCP and prosimian NTCP, respectively. The amplified product was then infusion cloned (Clontech, Mountain View CA) into the hNTCP lentiviral backbone vector which had been digested with XhoI/BsrGI following the manufacturer's instructions. For humanized mutant orthologues based on CynoNTCP (CynoNTCP$^{R158G}$, CynoNTCP$^{G157K,R158G}$, CynoNTCP$^{R158G,I160V}$, CynoNTCP$^{R158G,L161I}$, CynoNTCP$^{R158G,P165L}$, CynoNTCP$^{h157-165}$, CynoNTCP$^{Q84R, N86K}$ and CynoNTCP$^{Q84R,N86K+h157-165}$) and SqMNTCP (SqMNTCP$^{G158R}$, SqMNTCP$^{Q84R}$, SqMNTCP$^{K87N}$, SqMNTCP$^{N86K,K87N}$, SqMNTCP$^{Q84R,N86K,K87N}$), mutations were introduced to the parental constructs by QuickChangeXL Site-Directed Mutagenesis Kit (Agilent Technologies, Santa Clara, CA). The primers used for the cloning were listed in the Supplementary Table 2. These constructs were transformed into Stellar E. coli cells (Clontech, Mountain View CA). Colonies were picked, and grown in the presence of Amp selection, mini-prepped (Qiagen, Hilden, Germany) and confirmed by Sanger sequencing (Eton Bioscience, San Diego, CA).

### Generation of HepG2 cells expressing different NTCP orthologues
Lentiviral pseudo-particles were generated by co-transfecting 4.0E + 6 293T cells in a 10 cm tissue-culture plate using Xtremegene (Sigma-Aldrich, St. Louis, MO) with plasmids expressing the respective pLVX-NTCP-tagRFP proviral DNA, HIV-1 *gag–pol*, and VSV-G at a ratio of 1/0.8/0.2. Supernatants were collected at 24, 48 and 72 h, pooled, and filtered through a 0.45 μm filter (Millipore, Burlington, MA). Filtered

lentiviral supernatants were supplemented with polybrene (4 µg/mL, vol/vol) and (1:50, vol/vol) 1 M HEPES, aliquoted, and stored at −80 °C until use. Next, HepG2 cells were transduced with the human-, OWM- (with or without humanizing mutations), NWM- (with or without humanizing mutations), or prosimian-NTCP-tagRFP lentiviruses. After 3 days, expression of the fusion protein was assessed by both fluorescence microscopy using an EVOS microscope (Fisher Scientific, Waltham, MA) and LSRII Multi-Laser Analyzer (BD, Franklin Lakes, NJ) at the Princeton flow cytometry core facility. Each cell line generated showed greater than 90% of the cells expressing the respective fusion construct.

## Fibroblast culture
Primary marmoset fibroblasts were obtained from Coriell Institute (Camden, NJ). Fibroblasts once thawed were resuspended in a α-MEM (Thermo Fisher Scientific, Waltham, MA) supplemented with 10% FBS (Thermo Fisher Scientific, Waltham, MA), NEAA, GlutaMAX (both from Thermo Fisher Scientific), 1% Pen/Strep, and 64 mg/L L-ascorbic acid 2-phosphate sesquimagnesium salt hydrate (Sigma-Aldrich, St. Louis, MO). Cells were plated in a single well of a 6-well plate coated with 4 µg/cm² human fibronectin (Sigma-Aldrich). Cultures were grown at 5% $CO_2$/5% $O_2$ until confluent and then split using 0.05% trypsin. For routine passaging cells were cultured at 5% $CO_2$ and atmospheric oxygen.

## iPSC generation and differentiation of HLCs
Marmoset fibroblasts were grown at 5% $CO_2$/atmospheric $O_2$ in fibroblast media until 70–80% confluence. $1 \times 10^6$ cells were transfected with 1.5 µg per episomal vector containing the following genes: *OCT3/4, SHp53, SOX2, KLF4, LIN28*, and *L-MYC* (Addgene plasmids 27077, 27078, 27080 and 27082) and similar to as described[56] using the NEON Transfection System at (1,650 V, 10 ms, 3 time pulses). Transfected cells were seeded at 15,000/cm² on tissue culture plates with a CF-1 derived MEF feeder layer (Cornell University Stem Cell Core). Cells were grown in hESC media (made in house as previously described in[41]. At days 14–28, colonies were identified and detached via picking onto dishes coated with γ-irradiated CF-1 derived MEF and subsequently grown in hESC media (as described above) supplemented with 100 ng/ml human bFGF (Peprotech, East Windsor, NJ). Clones were routinely split using Rho-associated kinase (ROCK) inhibitor Y27632 (Tocris, Minneapolis, Minnesota) at a concentration of 10 µM. Cells were migrated to cell culture dishes precoated with 0.01 mg/cm² (1:100) of hESC-grade Matrigel (Corning, Corning, NY) and in mTESR media (Stem Cell Technologies, Vancouver, Canada). Undifferentiated iPSCs were maintained and differentiated into hepatocyte-like cells as previously described[41]. In brief, iPSCs were cultured in monolayer on Matrigel (Corning), and directed differentiation was achieved by sequential exposure to activin A, bone morphogenic protein 4 (BMP4) and basic fibroblast growth factor (bFGF), hepatocyte growth factor (HGF), and oncostatin M (OSM) (all cytokines from RND Systems, Minneapolis, MN).

## Functional characterization of iPS-MHLCs
Urea secretions were analyzed by measuring the concentration of urea in phenol-red free culture medium. The medium was collected and replaced with fresh medium every 2 days. The collected medium was centrifuged at 100 *g* for 5 min. The supernatant was stored at −20 °C for analysis. Urea concentration was assayed using a colorimetric endpoint assay using diacetylmonoxime with acid and heat (Stanbio Labs).

After 25 days of culture, hepatocyte-like cells were analyzed for cytochrome P450 activity. For enzyme induction experiments, hepatocyte-like cells were pretreated with inducers for 72 h. Stock solutions of inducers were prepared in dimethyl sulfoxide (DMSO) and diluted at 1:1000 for final concentrations of 50 mM omeprazole

(Sigma, St Louis, MO) and 25 mM rifampin (Sigma, St Louis, MO). Vehicle controls were pretreated with 72 h of 1:1000 DMSO. CYP450 activity was assessed with luminogenic and fluorogenic substrates for nonlytic assays using cultured cells. Briefly, hepatocyte-like cells were incubated with Luciferin-IPA (Promega, Madison, WI) (1:1000 dilution in phenol-free DMEM), Luciferin-H (CYP2C19, 1:50 dilution) or 7-benzyloxy-trifluoromethylcoumarin (BFC) (Multiple CYP450 isoforms 50 µM, Sigma) for 3 h. The spent medium was removed, and hepatocyte-like cells were washed with PBS 3 times. The hepatocyte-like cells were incubated with substrate-containing medium for 3 h at 37 °C and the medium was placed in Eppendorf tubes and frozen at −20 °C for further analysis. Metabolite conjugates formed from BFC were incubated with β-glucuronidase/arylsulfatase (Roche, Little Falls, NJ) for 2 h at 37 °C. Samples were diluted 1:1 in quenching solution and formation of metabolites was quantified with a fluorescence microplate reader (Molecular Devices, San Jose, CA) as described elsewhere[41]. Metabolite conjugates formed from Luciferin H and CYP3A4-IPA were processed and analyzed per Promega protocol and analyzed using a microplate luminometer (Molecular Devices, San Jose, CA).

## Construction of HBV/WMHBV chimeric virus clones
A chimeric HBV/WMHBV preS1[1–48] virus clone was constructed by QuickChangeXL Site-Directed Mutagenesis Kit (Agilent Technologies, Santa Clara, CA). Briefly, a pair of long chimeric primers (HBV-gtD-WMHBV preS1[1–48]-direct mutation-F, HBV-gtD-WMHBV preS1[1–48]-direct mutation-R) including WMHBV preS1[1–48] coding sequences and HBV flanking sequences were designed. By using HBV 1.3-mer gtD infectious clone as the templates, HBV/WMHBV preS1[1–48] virus clones were PCR amplified through QuickChangeXL Site-Directed Mutagenesis Kit (Agilent Technologies, Santa Clara, CA) following the manufacturer's instructions. The construct was confirmed by sequencing (Eton Bioscience, San Diego, CA). The primers used for the cloning are listed in Supplementary Table 3.

## Generation cell-culture-derived HBV and HBV/WMHBV chimeric virus
HBV infectious clones were constructed in our lab as previously described[57] and were used for producing HBV viral stocks. A WMHBV infectious clone (kindly provided by Robert Lanford, Texas Biomedical Research Institute) was used for producing WMHBV stocks. HBV/WMHBV chimeric virus stocks were produced using the constructed HBV/WMHBV chimeric clones. HepG2 cells were seeded into two 150 mm collagen coated plates at a density of 7.0E + 6 cells per plate. When the cell confluence reached about 90%, medium was changed to 25 mL DMEM/F12 (#11320082, Life Technologies, Carlsbad, CA) supplemented with 5% (vol/vol) fetal bovine serum (FBS), 1% (vol/vol) pen/strep (P/S) prior to the DNA transfection. Cells in each plate were transfected with 28 µg of plasmid DNA mixed with 2800 µL Opti-MEM™ Reduced Serum Medium (#51985034, Thermo Fisher Scientific, Waltham, MA) and 84 µL X-tremeGENE™ HP DNA Transfection Reagent (Sigma-Aldrich, # 6366546001, St. Louis, MO). About 4–6 h later, cells were washed three times with 25–30 mL pre-warmed DPBS and then cultured in 20 mL DMEM/F12 with 5% (vol/vol) FBS, 1% (vol/vol) P/S. Two hours later, cells were again washed three times with pre-warmed 25–30 mL DPBS. Cells were subsequently cultured in 20 mL DMEM/F12 with 5% (vol/vol) FBS, 1% (vol/vol) P/S. Supernatants were harvested every 2–3 days for a total of 10–11 days and stored at 4 ˚C temporarily. Supernatants were either spun at 3000 *g* for 15 min at 4 ˚C or passed through a 0.45 µm filter (Thermo Fisher, Waltham, MA) to remove cell debris. Then, supernatants were transferred and concentrated either by polyethylene glycol (PEG) 8,000 (Sigma-Aldrich, St. Louis, MO) precipitation or using heparin columns (GE Healthcare, Chicago, IL). For PEG 8000 concentration, viruses were concentrated by precipitation using 8% (vol/vol) PEG 8,000, shaking at 15 rpm overnight at

4 ˚C. The viruses were pelleted at $4200\,g$ for 1 h and 15 min at 4 ˚C. Supernatants were removed carefully and the precipitate was resuspended in 1% (vol/vol) of the original supernatant volume in PBS supplemented with 10% (vol/vol) FBS on a shaker at 25 rpm, overnight at 4 ˚C. The viruses were then spun at $800\,g$ for 8 min at 4 ˚C. The supernatant was transferred into a sterile tube and aliquoted. For the heparin column concentration and purification, supernatants were run through the 5 mL heparin columns at a speed of 5 mL/min. Then the columns were washed once with 8 mL 1×PBS followed by elution with 30 mL elution buffer (25 mL 10×PBS pH 7.4 diluted into 100 mL with 75 mL distilled water). The elution was collected and further concentrated with the centrifugal filters (100,000 NMWL, Merck Millipore Ltd.) at $3000\,g$ for 30 min at 4 ˚C. Finally, about 1.6 mL virus stocks were transferred and aliquoted in LoBind Microcentrifuge Tubes (Eppendorf). Then 20 µL of virus stock was used and DNase I was added (37 ˚C, 1 h) to digest possible residual plasmid DNA. DNA was extracted with the QIAmp MinElute virus spin kit (Qiagen, Hilden, Germany). Thereafter, HBV titers were quantified by qPCR. Based on the HBV DNA copy numbers virus was aliquoted and cryopreserved at −80 °C until use.

### Generation of HBV 1.3-mer lentivirus

To enable HBV replication and viral protein expression under HBV endogenous enhancers and promoters, the CMV promoter was removed from the pLVX-CMV-IRES-Puro lentiviral vector by Site-Directed Mutagenesis (Agilent Technologies, Santa Clara, CA). The pLVX$^{\Delta CMV}$-IRES-Puro lentiviral vector was then linearized by ClaI (#RO197, New England Biolab, NEB, Ipswich, MA) and BamHI (#RO136, NEB). HBV 1.3-mer genome (gtD, awy) fragment was then amplified by using HBV 1.3-mer infectious clone as PCR template and PCR primers with ClaI and BamHI flanking sequences, respectively, which was constructed in our lab as previously described[57]. The linearized vector backbone and the HBV 1.3-mer genome were then ligated by infusion cloning per the manufacturer's instructions (Clontech, Mountain View CA), and were transformed into Stellar *E. coli* cells (Clontech, Mountain View CA). Colonies were picked, and grown in the presence of Amp selection, mini-prepped (Qiagen, Hilden, Germany) and confirmed by Sanger sequencing (Eton Bioscience, San Diego, CA).

Lentiviral pseudo-particles were generated by co-transfecting $4 \times 10^6$ 293 T cells in a 10 cm plate using Xtremegene (Sigma-Aldrich, St. Louis, MO) with plasmids expressing the pLVX$^{\Delta CMV}$–1.3-mer-HBV-IRES-Puro proviral DNA, HIV-1 *gag–pol*, and VSV-G in a ratio of 1/0.8/0.2. Supernatants were collected at 24, 48 and 72 h, pooled, and filtered through a 0.45 µm filter (Millipore, Burlington, MA). Filtered lentiviral supernatants were supplemented with polybrene (4 µg/mL final concentration) and (1:50, vol/vol) 1 M HEPES, aliquoted, and stored at −80 °C until use. The primers used for the cloning are listed in Supplementary Table 4.

### Transduction of SACC-PMHs with HBV 1.3-mer or hNTCP lentivirus

10 days following the establishment of the SACC-PMH culture, cells were transduced with 150 µl of undiluted HBV 1.3-mer lentivirus, hNTCP lentivirus or mock lentivirus as control, incubated overnight, and washed twice with Hμrel maintenance medium™. For SACC-PMHs transduced with the HBV 1.3-mer lentivirus, cultures were maintained in 100 µl/well Hμrel maintenance medium™, and the supernatants were collected every two days for the HBV infection assay. For SACC-PMHs transduced with hNTCP lentivirus, cells were pretreated and infected with HBV in the same manner as naïve SACC-PMHs.

### HBV infection of human hepatoma cells

HBV infections of HepG2 cells overexpressing different NTCP orthologues and un-transduced HepG2 cells were performed as follows: Naïve HepG2 or HepG2-NTCP cells were seeded into 48-well plates or 24-well plates at 5.0E + 4 or 1.0E + 5 cells/well, respectively, and then were pre-treated with pretreat medium (DMEM supplemented with 3% (vol/vol) FBS, 2% (vol/vol) DMSO, 1% (vol/vol) Pen/Strep) for 24 h. HBV infections with cell-culture derived HBV, WMHBV or HBV/WMHBV preS1[1–48] chimeric virus stocks were used at a MOI of 8,000 unless indicated otherwise, in the presence of 4% polyethylene glycol (PEG) 8,000 (Sigma-Aldrich, St. Louis, MO) and 2% dimethyl sulfoxide (DMSO, Sigma-Aldrich, St. Louis, MO). After inoculating the cells with viruses for 12–24 h, the cells were washed with pre-warmed sterile 1×PBS for at least 4 times and then incubated with fresh maintenance media (DMEM supplemented with 3% FBS, 2% DMSO, 1% Pen/Strep and 1x non-essential amino acids). For the blocking assay, media supplemented with 750 nM Mycludex B (kindly provided by Stephan Urban, University of Heidelberg) was used for the HBV infections and the drug was kept continuously in the duration of the experiment. Samples were collected at the indicated time-points.

### HBV infection of primary marmoset hepatocytes in the SACC-PMH system

10 days following the establishment of the SACC-PMH culture, cells were pretreated with 100 µl Hμrel maintenance medium™ (Visikol Inc., Hampton, NJ) supplemented with 0.5% DMSO for 24 h. HBV infections with cell-culture derived HBV, WMHBV or HBV/WMHBV preS1[1–48] chimeric virus stocks were used at a MOI of 8,000 unless indicated otherwise, in the presence of 4% polyethylene glycol (PEG) 8,000 (Sigma-Aldrich, St. Louis, MO) and 0.5% dimethylsulfoxid (DMSO, Sigma-Aldrich, St. Louis, MO). The cells were then washed four times with pre-warmed Hμrel PlatinumHeps maintenance medium™ supplemented with 0.5% DMSO and then maintained with 100 µl Hμrel maintenance medium™ supplemented with 0.5% DMSO. Cell morphology was monitored under bright field light using an EVOS microscope (Fisher Scientific, Waltham, MA). The medium was harvested and replaced every 2 days.

### Quantification of HBsAg and HBeAg

Detection and quantification of HBsAg and HBeAg levels were performed by a chemiluminescence immunoassay (CLIA) according to the manufacturer's instructions (Autobio Diagnostics CO., LTD, Zhengzhou, Henan, China) or the HBsAg levels were tested by the HBsAg EIA 3.0 kit (Bio-Rad, Hercules, CA). For samples from sucrose density gradient centrifugation, a 50 µl sample of a 1:10 dilution in 1× sterile PBS was used. For samples derived from infection experiments, 50 µl supernatant was processed for HBsAg and HBeAg quantification directly.

### Isolation of HBV DNA and RNA from HepG2 cells, PMHs and iPS-MHLCs

Total DNA and RNA from infected cells was extracted by using a Quick-DNA/RNA Microprep Plus Kit (Zymo Research, Irvine, CA) following the manufacturer's instructions. Briefly, cells were resuspended in 300 µl DNA/RNA Shield and then digested with 15 µl Proteinase K (20 mg/ml) for 30 min. 300 µl DNA/RNA lysis buffer was subsequently added and mixed. The sample solution was then transferred into the Zymo-Spin IC-XM column to harvest the DNA. The flow-through from the last step was saved and an equal volume of ethanol was added to purify RNA by using the Zymo-Spin IC column. Finally, the DNA/RNA was then eluted with 30 µl of nuclease-free water and concentrations were measured using a Nanodrop spectrophotometer (Thermo Fischer Scientific, Waltham, MA).

### Quantification of HBV DNA by quantitative PCR

A 2 µl aliquot of HBV DNA isolated either from supernatants after transfection or infection was used per reaction well. We used a well-characterized HBV rcDNA qPCR system with HBV-qF (nt 1776–1797, numbered based on gt D with GenBank accession no. U95551.1): 5′-

GGAGGCTGTAGGCATAAATTGG-3', HBV-qR (nt 1881-1862, numbered based on gt D with GenBank accession no. U95551.1): 5'-CACAGCTTGGAGGCTTGAAC-3' covering the conserved region of HBV(LLD ≈ 1.0E + 3 copies/mL) as described previously[58]. Primers were kept at a final concentration of 500 nM in a 20 μl reaction volume. The following PCR program was run on a Step One Plus qPCR machine (Life Technologies): denature 95 °C for 10 min, followed by 40 cycles of 95 °C for 30 s, 60 °C for 30 s, and 72 °C for 25 s.

## Quantification of HBV cccDNA by quantitative PCR

HBV infection was performed in 12-well-plates in which 4.0E + 5 HepG2-marmoset-NTCP cells were seeded per well and the infections were carried out similarly as for HBV infection of human hepatoma cells. Five days after infection, Hirt DNA extraction was performed as previously described[39]. Briefly, cells were lysed by adding 400 μl Hirt DNA lysis buffer (10 mM Tris-HCl, pH 8.0; 0.625% SDS; 10 mM EDTA) with gentle shaking at room temperature for 30 min. The cell lysates were then transferred to 2 mL tubes followed by adding 100 μL of 5 M NaCl. Cell lysates were then mixed and incubated overnight at 4 °C followed by a spin at 12,000 g for 30 min at 4 °C. The supernatants were transferred to new centrifuge tubes into which an equal amount of phenol was added. The samples were mixed thoroughly and spun at 12000 g for 10 min at 4 °C. These steps were repeated once before adding an equal amount of phenol-chloroform (1:1) to extract the DNA followed by isopropanol precipitation and 70% ethanol wash. The Hirt DNA was then dissolved in 10 μL nuclease-free water and the samples from 2 wells for each group were combined. Then 5 μL Hirt DNA was used for T5 treatment (#M0663S, NEB, Ipswich, MA) for 30 min at 37 °C in a 10 μL reaction system with 0.5 μL T5 exonuclease followed by heat-inactivation at 95 °C for 5 min. Then the samples were diluted with 30 μL nuclease-free water and 4 μL samples from each group were used for HBV cccDNA specific qPCR. To detect HBV cccDNA, cccDNA-F(5'-GCCTATTGATTG-GAAAGTATGT-3') and cccDNA-R (5'-AGCTGAGGCGGTATCTA-3') primers were used with a program of 95 °C for 10 min, 45 cycles of 95 °C 15 s, 60 °C 5 s, 72 °C 45 s and 88 °C 2 s for fluorescence signal acquisition as previously described[40]. A 1.3 × HBV plasmid constructed previously in our lab[57] was used to generate a standard curve to calculate HBV cccDNA copy number. For infection on PMHs, total DNA was extracted and a T5 exonuclease digestion was performed, and the HBV cccDNA quantification was same as above and previous studies[39,40].

## Quantification of HBV pgRNA by quantitative RT-PCR

HBV RNA purified from supernatants was used to determine extracellular nucleocapsid-associated HBV pre-genome RNA (pgRNA) level as described previously[59]. Briefly, 7.5 μl of isolated nucleic acid was treated by DNase I (Thermo Fisher Scientific, Waltham, MA, USA) followed by reverse transcription with a specific HBV primer (5'-CGAGATTGA-GATCTTCTGCGAC-3', nt 2415–2436, numbered based on gt D with GenBank accession no. U95551.1) located in precure/core region[59] using RevertAid™ First Strand DNA Synthesis Kit (Thermo Fisher Scientific, Waltham, MA, USA). For quantitative assays, the standards with 1-mer HBV target template were cloned into the TOPO-Blunt Cloning vector (Thermo Fisher, Waltham, MA, USA #450245) and the copy number was calculated based on the vector molecular weight and concentration. A master mix was created containing 15 μl 2 × Taqman reaction mix (Applied Biosystems, Waltham, MA, USA), 500 nM forward and reverse primers, 200 nM probe and 3 μl synthesized cDNA in a 30 μL reaction. The master mix was then added to the samples or to the 10-fold serial dilution standards and the following cycling program was used to run the qPCR: 95 °C for 10 min; 45 cycles of 95 °C 15 sec and 58 °C for 45 sec.

## Southern Blot

HBV DNA detection by Southern blotting was carried out by using the DIG labeled probe. Briefly, 1.0 × HBV genome (gt D) plasmid was constructed as the template by using the HBV full length primers:

HBV-FL-F:    CCGGAAAGCTTGAGCTCTTCTTTTTCACCTCTGCC-TAATCA; HBV-FL-R:    CCGGAAAGCTTGAGCTCTTCAAAAAGTTG-CATGGTGCTGG as previously described[57]. Then using this plasmid, an HBV full length DNA probe was generated by the PCR DIG Probe Synthesis Kit (Roche #11636090910, MilliporeSigma, Rockville, MD). Accordingly, 3.2 kb and 2.0 kb HBV DNA markers were amplified by regular PCR using the HBV full length primers above and HBV-FL-F and HBV-2.0-R (TGAGGCCCACTCCCATAGG), respectively. HBV infection was performed in 12-well-plates in which 4.0E + 5 HepG2-marmoset-NTCP cells were seeded per well and infections were carried out similarily as those for HBV infection of human hepatoma cells. Five days after infection, cells were lyzed by adding 400 μl lysis buffer (10 mM Tris-HCl, pH 8.0; 0.625% SDS; 10 mM EDTA) and shaking at room temperature for 30 min. The cell lysates were then transferred to 2 mL tubes and 200 μl 3 × Proteinase K buffer (15 mM Tris-HCl pH 8; 300 mM NaCl; 3 mM EDTA; 1.5% SDS) together with 2 μl 25 mg/mL Proteinase K (MilliporeSigma, Rockville, MD) was added. The mixtures were then incubated in a Thermomixer (Eppendorf) at 55 °C, 300 rpm for 2 h followed by a standard Phenol/Chloroform purification. After isopropanol precipitation and 70% ethanol wash, the DNA pellets were dissolved with 30 μl distilled water. Then the DNA samples from 4 wells of mock, HBV and the chimeric WMHBV/HBV infection groups were pooled, respectively and then the total HBV DNA was purified through the MinElute PCR Purification Kit (Qiagen, Hilden, Germany) which can selectively purify DNA fragments from 70 bp ~ 4 kb. The DNA from each group were then resolved on a 1.3% (wt/vol) agarose gel. The gel was treated with denaturing buffer (0.5 M NaOH, 1.5 M NaCl) for 2 × 15 min at room temperature, with gentle shaking. The gel was rinsed with sterile, doubly distilled water briefly and then submerged into the neutralization buffer (1 M Tris-HCl pH 7.5, 1.5 M NaCl) for 2 × 15 min at room temperature, with gentle shaking. After equilibrating the gel in the standard 20 × SSC buffer (Roche #11666681001) for 10 min, the DNA was then transferred onto a positively charged nylon membrane (Roche #11209272001) using Whatman 3MM paper (#3030-866, GE Healthcare, Chicago, IL) and 20 × SSC buffer by overnight capillary transfer. The DNA was then cross-linked to the membrane by UV irradiation at 120,000 μJ for 2 min (UV Crosslinker, Model 13-245-221, Fisher Scientific). The membrane was washed briefly in 2 × SSC for 5 min and allowed to air dry. The membrane was then incubated with the DIG Easy Hyb (Roche # 11603558001) for 1 h at 55 °C, before hybridization with DIG-labeled HBV full length probe at 55 °C overnight. The hybridized membrane was washed twice with washing buffer (2 × SSC, 0.1% SDS) for 5 min at room temperature followed by 2 × 20 min washes with stringent washing buffer (2 × SSC, 0.1% SDS) at 60 °C and was subsequently rinsed briefly in maleic acid buffer (Roche #11585762001) and blocked at room temperature for 30 min in blocking buffer (Roche #11585762001). Then the membrane was incubated with alkaline phosphatase-conjugated DIG antibody (1:10,000; Roche #11093274910) for 30 min at room temperature followed by 3 × 15 min washing with washing buffer (Roche #11585762001). After equilibration in detection buffer (Roche #11585762001) for 3 min, the membrane was subjected to chemiluminescent detection using the CDP-Star solution (1:500; Roche #11759051001) and the signal was captured through the chemiluminescence channel on a blot Imaging machine (iBright 1500, Invitrogen, Waltham, MA).

## Multiplexed HBV RNA ISH, HBc and NTCP IHC assay

HBV infection was performed in a 12-well-plate in which 4.0E + 5 cells were seeded per well and the infection protocol was the same as that for HBV infection of human hepatoma cells. Five days after infection, cells were detached using 0.05% trypsin and the cells from 17 wells of each group were combined and spun at 1,000 rpm for 10 min. The cell pellets were washed once with 1 × PBS, then cells were re-suspended in about 2 ml of 10% (vol/vol) formalin and incubated 20 min at room

temperature. After the spin, the cells were washed once with $1 \times$ PBS and then were resuspended in 1 mL $1 \times$ PBS and stored at 4 °C until proceeding to Histogel preparation. In addition to the test samples, HepG2-marmoset-NTCP mock control and HepG2 cells transfected with 1.3xHBV plasmid control and naive HepG2 cells acting as the double negative cell pellets were also prepared. Cell pellets were embedded using Histogel (Kalamazoo, MI, USA) prior to being processed via a Tissue-Tek VIP-5 automated vacuum infiltration processor (Sakura Finetek USA, Torrance, CA, USA), followed by paraffin embedding using a HistoCore Arcadia paraffin embedding station (Leica, Wetzlar, Germany) to generate formalin-fixed, paraffin-embedded (FFPE) cell blocks. Samples were sectioned to 5 μm, transferred to positively charged slides, deparaffinized in xylene, and dehydrated in graded ethanol. Tissue staining was conducted using a Ventana Discovery Ultra (Roche, Basel, Switzerland). Details for the combined ISH-IHC assay are outlined in Supplemental Table 5. ISH probes targeting HBV (genotype D, ayw) S and X gene regions were designed and manufactured by Molecular instruments (#LSA204, Los Angeles, CA, USA). Note, Molecular instrument does not currently allow publication of the individual probe binding sequences, but identical probes are available per request. In addition to an HBV RNA probe for test samples, an internal housekeeping probe (UBC, #N451GP, Los Angeles, CA, USA) and a *Bacillus subtilis* negative control (dapB, #H307GP, Los Angeles, CA, USA) probe were employed to ensure RNA integrity and specificity respectively. Probes were hybridized for 2 h at 42 °C following the manufacturer's instructions and developed using a DAB chromogen (Roche). Following completion of the ISH sequence, horse radish peroxidase (HRP) was inhibited using an endogenous peroxidase blocker and the IHC sequences were conducted following primary and secondary HRP polymer (Vector Laboratories, Burlingame, CA) antibody incubations. Heat stripping was conducted after the first purple (Roche) IHC sequence to remove primary and secondary antibody complexes prior to the development of the second yellow (Roche) IHC reaction. All positive and negative controls were confirmed ensuring specificity of the assay. For figure preparation, 600x representative images were taken or 400× total magnification digitalized whole slide images (WSI) of the multiplex ISH-IHC combination assay were generated using a multispectral Vectra Polaris Automated Quantitative Pathology Image System (Akoya Biosciences, Marlborough, MA, USA).

## HBV/WMHBV preS1 peptide binding and competitive inhibition assays

N-terminal myristoylated peptides with amino acids 2–48 of HBV (genotype D, GeneBank U95551.1) or WMHBV preS1 (GeneBank AY226578.1) conjugated with FITC at the C-terminal were synthesized by GenScript USA Inc. (Piscataway, NJ). The peptides were dissolved in dimethyl sulfoxide (DMSO, Fisher Scientific, Waltham, MA) to generate the stock solutions. For the binding assays, 3.0E + 4 HepG2 cells or HepG2 cells expressing different NTCPs were seeded in each well of the round-bottom 96-well plates and were cultured in 100 μL pretreatment medium (3% (vol/vol) FBS, 2% (vol/vol) DMSO, 1% (vol/vol) Pen/Strep and $1 \times$ NEAA in DMEM medium) with 0, 50, 100 and 200 nM peptides, respectively. For competitively inhibition assays, 200 nM peptides and 750 nM Myrcludex B (MyrB, kindly provided by Stephan Urban, University of Heidelberg) were included in the medium. After 1 h incubation in 37 °C, unbounded peptides were washed out by FACS buffer (PBS with 1% (vol/vol) FBS) followed by 4% (vol/vol) PFA fixing at 4 °C for 30 min. The cells were then washed for another two times and resuspended in 200 μl FACS buffer. NTCP expression and HBV/WMHBV binding were analyzed by flow cytometry to determine tagRFP expression and FITC signal, respectively using a LSRII Multi-Laser Analyzer (BD, Franklin Lakes, NJ) at the Princeton flow cytometry core facility.

To characterize the binding by imaging, 8.5E + 4 HepG2 cells or HepG2 cells expressing different NTCPs were seeded in each well of 24-well plates with a collagen precoated cover slide and were cultured in 500 μL pretreatment medium (3% (vol/vol) FBS, 2% (vol/vol) DMSO, 1% (vol/vol) Pen/Strep and 1X NEAA in DMEM medium) with 200 nM peptides at 37 °C for 1 h. The cells were then washed twice and fixed with 4% (vol/vol) paraformaldehyde (PFA) at 4 °C for 30 min followed by washing two more times. DAPI (#D9542, MilliporeSigma, Rockville, MD) was used to stain nuclei (1:1000 dilution, 1 μg/mL), incubated with cells for 15 min at room temperature followed by washing twice. The cover slides with cells were then transferred onto a glass slide with 2 μl ProLong Gold antifade solution (#P36930, Thermo Fisher Scientific, Waltham, MA) and sealed with CoverGrip™ Coverslip Sealant (#23005, Biotium Inc., Fremont, CA). The slides were then covered with foil and dried overnight at 4 °C and then analyzed with a confocal microscope (Nikon A1R-STED) in the Imaging Core facility at Princeton University.

## Virus binding and entry assays

7.0E + 4 HepG2 or HepG2-NTCP cells were seeded into each well of 48-well plates and then cultured with 300 μl pretreatment medium (DMEM, 2% DMSO, 3% FBS, 1% P/S, vol/vol) for 24 h. Prior to virus inoculation, cells were placed on wet ice for 15 min, then exposed to prechilled HBV, WMHBV or HBV/WMHBV preS1[1–48] inoculum in the presence of 4% (w/vol) polyethylene glycol (PEG) 8,000 (Sigma-Aldrich, St. Louis, MO) and 2% (vol/vol) dimethylsulfoxid (DMSO, Sigma-Aldrich, St. Louis, MO) and incubated for another 15 min on ice to allow for viral attachment. For the binding assay, the cultures were kept at 4 °C for 1 h followed by 4 washes with 500 μl/well cold PBS to remove unbound particles. The cells were then lysed using 400 μl lysis buffer (10 mM Tris-HCl, pH 8.0; 0.625% SDS; 10 mM EDTA). For the entry assay, cultures were shifted to 37 °C following viral attachment to the cells, incubated for 16 h and then washed once by using 500 μl/well room temperature $1 \times$ PBS. The cells were then trypsinized for 3 min with 150 μl Trypsin (0.05%, diluted from 0.25% Trypsin by Ca⁺-, Mg⁺-free PBSS,), then quenched by adding 150 μl regular DMEM culture medium (10% FBS, 1xP/S). After centrifugation the medium was aspirated, and cells were washed again with 500 μL $1 \times$ PBS. The cells were then lysed using 400 μl lysis buffer (10 mM Tris-HCl, pH 8.0; 0.625% SDS; 10 mM EDTA). HBV DNA was isolated from the lysates by standard Phenol/Chloroform extraction and quantified by qPCR.

## Immunofluorescence of HBc

HepG2 cells +/- NTCP were grown on collagen-precoated cover slides and fixed 6 days after infection with 4% (vol/vol) paraformaldehyde (PFA). The cells were then washed 3 times with $1 \times$ PBS followed by 1 h of incubation with the blocking and permeabilization buffer (5% BSA, 5% FBS, 0.3% Triton-X-100 in $1 \times$ PBS, all (vol/vol)) at room temperature. Thereafter, cells were incubated with an anti-HBc antibody (1:200 dilution, #LS-C67477, LSBio, Seattle, WA) or an anti-HBc antibody (1:500 dilution, #B0586, Dako, Denmark) in the dilution buffer (1% BSA, 1% FBS, 0.3% Triton-X-100 in $1 \times$ PBS) at 4 °C overnight, washed for 3 times and incubated with an Alexa Fluor® 488 conjugated goat anti-mouse secondary antibody (for LSBio antibody, 1:1000 dilution, #A28175, Thermo Fisher Scientific, Waltham, MA) or Alexa Fluor® 488 conjugated goat anti-mouse secondary antibody (for Dako antibody, 1:1000 dilution, #A11008, Thermo Fisher Scientific, Waltham, MA) and DAPI (1 μg/mL) in dilution buffer (1% BSA, 1% FBS, 0.3% Triton-X-100 in $1 \times$ PBS) at room temperature for 1 h. After 3 times of washing, the cover slides with cells were then transferred onto a glass slide with 2 μl ProLong Gold antifade solution (#P36930, Thermo Fisher Scientific, Waltham, MA) and sealed with CoverGrip™ Coverslip Sealant (#23005, Biotium Inc., Fremont, CA). The slides were then covered with foil and dried overnight at 4 °C and then analyzed with a confocal microscope (Nikon A1R-STED) in the Imaging Core facility at Princeton University.

The images were analyzed and exported by ImageJ (Version: 2.1.0/1.53c).

## Sucrose density gradient centrifugation

20 mL supernatant was collected from day 2 to day 4 post transfection of WMHBV, HBV or HBV/WMHBV preS1[1–48] infectious clones into HepG2 cells. The supernatants were then centrifuged in 50-mL tubes at 1700 g for 15 min and approximately 19 mL of cleared supernatant were carefully layered onto a discontinuous sucrose gradient from 60–15% (w/w, 2 mL of 60%, 4 mL of 45%, 4 mL of 35%, 5 mL of 25%, 5 mL of 15%)) in TNE buffer (20 mM Tris-HCl, pH 7.4, 140 mM NaCl, 1 mM EDTA) in a 38.5 mL polyallomer ultracentrifuge tube (Beckman Coulter, Inc., Indianapolis, IN). The gradient was allowed to rest for at least 4 h at 4 °C before proceeding with centrifugation in a SW 28 rotor at 25,000 rpm (112,000 g) for 15 h at 10 °C in an Optima XE-100 ultracentrifuge (Beckman Coulter, Inc., Indianapolis, IN). Fractions (1.3 mL each) were collected from bottom to top. The sucrose density in different fractions was determined with a refractometer (Hogentogler & Co. Inc., Columbia, MD) and viral DNA was quantified by qPCR.

## Transmission electron microscopy (TEM)

For transmission electron microscopic analysis, 2 µl of sample from fraction 6 of the sucrose density gradient was placed on a glow-discharged carbon-coated 400-mesh copper grid (CF400-Cu-50, Electron Microscopy Sciences, Hatfield, PA)) followed by negative staining with uranyl formate (1%, pH 6.8) for 30 s, and then examined at 80 kV using a transmission electron microscope (Philips CM100 TEM) equipped with CCD camera in the Imaging and Analysis Center (IAC) of Princeton University.

## Phylogenetic analysis of NTCPs

The phylogenic trees were built by MegAlign Pro (DNASTAR Inc., Madison, WI) using a RAXML alignment of orthologous amino acid sequences of NTCPs from 38 species including 28 primates (Supplementary Table 1). The residue numbering is based on human NTCP. Tupaia is included in the assay as it shows near evolution relationship with primates and support HBV infection.

## NTCP structure prediction, molecular graphics and analysis

FASTA files of NTCP (WT or mutants) of each species were submitted to the AlphaFold algorithm[33] (Version 2.1.2, DeepMind, United Kingdom) run on the Princeton Research Computing DELLA Cluster at Princeton University. 5 models of each protein prediction were produced, and the best ranked model for each was used for subsequent analysis. Molecular graphics and analyses performed with Protein 3D (DNASTAR Inc., Madison, WI). Briefly, NTCP structures were shown and rendered either in Ribbon (cartoon) or molecular surface (solid or mesh) models. The side chains were visualized in some regions in ball and stick model to highlight important residues.

## Atomic distance calculation and multiple structure alignment

Atomic distance calculations between the NTCP structure predications produced via AlphaFold[33] (WT or mutants) and known human Cryo-EM NTCP structure (PDB:7PQG and 7PQQ, the nanobody was removed) were done via the structure-based alignment tool TM-Align[60] (Version 20190822). Briefly, the known crystal structure and the AlphaFold predications were fed into the TM-Align software. Since the structures of highly flexible regions at the C and N termini of NTCP could not be resolved by CryoEM but Alphafold produced structural predictions for these regions the amino acid sequences not present in the Cryo-EM human NTCP structure (AAs 12–310) were removed utilizing Protein 3D (DNASTAR Inc., Madison, WI). To enable direct structural comparisons different NTCP structures or specific regions were aligned using Protein 3D, and the root mean squared deviation (RMSD) was calculated to indicate the structural similarity.

## Statistics analysis

Bar graphs were presented as mean ± SEM with individual data points and experiments were repeated at least 2 times with duplicates or triplicates unless indicated otherwise. All data were analyzed with Prism v9.2.0 (GraphPad) and either Student's $t$ test (unpaired, two-tailed), nested $t$ test (two-tailed) or one-way ANOVA was used for the statistics. Statistical significance of $p$ values are indicated.

## Biosafety statement

Experiments described here have been approved by the Princeton University Institutional Biosafety Committee (# 1145). All work involving infectious HBV and WMHBV regents, including recombinant HBV/WMHBV preS1[1–48] strain, was performed in Biosafety Level 2 laboratory. All personnel working with the virus were trained with relevant safety and procedure-specific protocols. For the adapted HBV/WMHBV preS1[1–48] strain, the replaced region localizes at the beginning of HBV preS1 protein and the currently utilized HBV vaccines target antigenic epitopes within HBV S protein (downstream of preS1 specific region), so the antigenic epitopes are same between HBV/WMHBV preS1[1–48] and HBV and thus the adapted HBV/WMHBV preS1[1–48] strain should not result in immune escape from commonly used vaccines.

## Reporting summary

Further information on research design is available in the Nature Portfolio Reporting Summary linked to this article.

# Data availability

Data are available in the main text or the supplementary materials. Source data are provided with this paper. Human NTCP CDS, NCBI accession number: JQ814895.1; CynoNTCP CDS, NCBI accession number: AK240620.1; SqMNTCP CDS, NCBI accession number: XM_003924480.3; Marmoset NTCP CDS, NCBI accession number: XM_035260831.1; Prosimian NTCP CDS, NCBI accession number: XM_012656155.1; Human Cryo-EM NTCP structure, PDB:7PQG and 7PQQ. The AlphaFold related structure data have been deposited[61] in Zenodo with the access link https://zenodo.org/record/7799411#.ZC16cHvMJD8. Source data are provided with this paper.

# Code availability

The script used to generate the AlphaFold prediction has been deposited in Zenodo with the access link https://zenodo.org/record/7799411#.ZC16cHvMJD8.

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

## Acknowledgements

The authors thank Dr. Robert Lanford (Texas Biomedical Research Institute) for providing the WMHBV infectious clone and Dr. Stephan Urban (University of Heidelberg) for providing the Myrcludex B. We would like to thank Christina DeCoste and Katherine Rittenbach in the Molecular Biology Flow Cytometry core facility and Dr. Gary Laevsky and the Molecular Biology Confocal Microscopy Facility which is a Nikon Center of Excellence for their excellent technical support. We further thank all members of the Ploss lab, especially Dr. Robert LeDesma for critical discussions and comments throughout experimentation and preparation of the manuscript. Work in the lab is supported by grants from the National Institutes of Health (R01 AI138797, R01 AI107301, R01 AI146917, R01 AI153236 to A.P.), (R01AI107301, R01DK121072 to R.E.S.), a Burroughs Welcome Fund Award for Investigators in Pathogenesis (#101539 to A.P.) and funding from Princeton University. The Molecular Biology Flow Cytometry Resource Facility is partially supported by the Cancer Institute of New Jersey Cancer Center Support grant (P30CA072720). This work utilized NIH S10 Shared Instrumentation Grants S10-OD026983 and SS10-OD030269 (awarded to N.A.C.).

## Author contributions

This project was conceived by Y.L. and A.P. Bioinformatic analyses were performed by Y.L. and A.B. All other experiments were performed by Y.L., D.P., T.C., B.Y.W., Y.B., V.C. A.K.O., H.P.G., N.A.C., and R.E.S. C.C. generated the SACC-PMHs. All data were analyzed by Y.L. and A.P. The draft of the manuscript was prepared by Y.L and A.P. with edits from all authors.

## Competing interests

Y.L. and A.P. have filed a patent application on the generation and use of simian-tropic HBV for modeling HBV in vivo. Claims cover the use of HBV and stHBV of multiple genotypes to infect marmosets. There are no restrictions on the publication of the data. The remaining authors declare no competing interests.
