## [Peer Review File · Nature Communications]

Targeted viral adaptation generates a simian-tropic hepatitis B virus that infects marmoset cellsEditorial Note: Parts of this Peer Review File have been redacted as indicated to remove third-party material where no permission to publish could be obtained.

REVIEWER COMMENTS

Reviewer #1 (Remarks to the Author):

The authors examined the ability of NTCP derived from OWM, NWM, and prosimian as well as human for supporting HBV preS1 binding and infection. They further examined the mutant NTCPs to show that HBV infection was reproduced by R158G and further promoted by R158G/P165L in OWM-derived NTCP, and was observed by either Q84R, K87N, or additional combined mutations in SqM-NTCP. Structural prediction suggested that the mutation R158G/P165L in OWM NTCP changed the stability of the loop between TM4-5, which might be related with the elevated HBV infection. They showed the infection of WMHBV in primary marmoset hepatocytes. The HBV/WMHBVpreS1(1-48), a chimeric virus of which preS1(2-48) of HBV was replaced by that of WMHBV, was shown to infect HepG2-marmoset NTCP cells, primary marmoset hepatocytes, and iPS-derived hepatocyte-like cells more efficiently than HBV.

This information is important for establishing an in vivo surrogate infection model of HBV. The authors showed a lot of data for infection susceptibility of NTCP derived from different species and their mutants, which helps to understand the regulation mechanism of hepadnavirus entry. However, there are multiple inconsistent results between figures or those not well explained. The conclusion of this study is not clear. If the authors want to claim that the HBV/WMHBV preS1 chimera is useful for in vivo hepadnavirus infection model, they should show in vivo infection of this chimeric virus to marmosets. If the authors focused on the amino acids in NWM- or OWM-NTCP that determine the HBV susceptibility, majority of the results are already reported, including the importance of aa 158 of NTCP. The story of the paper has to be totally reconstructed to claim the significance of the priority of this paper. Other comments are shown below.

- In Fig. 2, the authors showed that SqM NTCP as well as OWM NTCP(R158G) supported HBV preS1 binding. Why SqM NTCP (WT) did not render HBV infection while OWM NTCP(R158G) supported a low level of HBV infection? Also, substitutions Q84R, K87N, N86K/K87N, and Q84R/N86K/K87N in SqM NTCP presented HBV infection. Do these mutations facilitate the HBV preS1 binding compared with the wild type SqM NTCP? The authors should present data to make the relationship between the HBV preS1 binding and HBV infection clearer.
- In Fig. 3, the authors speculated that the substitution at aa 165 in NTCP affect the structure of the TM4-5 region. The authors should at first mention the possibility that P165L in OWM NTCP may directly augment the binding of this site to HBV preS1 and examine this possibility. And then about the possibility the authors proposed in Fig. 3B, they should introduce mutations in the TM4-5 region to disrupt the structure seen for OWM (R158G/P165L) to examine whether the proposed hypothesis is true or not.
- In Fig. 3C, the reviewer doesn't understand how SqM NTCP(K87N) and marmoset NTCP are structurally more similar to human NTCP. The authors should show a quantitative index.
- Fig. 3D calculated RMSD of NTCPs. Since only a part of regions in NTCP, not the entire region, is involved in preS1 binding, it would be nice to present the similar analysis using the 84-87 and 157-165 aa and the surrounding regions, instead of using the entire NTCP.
- Fig. 4B should show images displaying several cells, not a single cell, to avoid unexpected bias when selecting the images.
- In Fig. 4F, why non-transfected PMH producing endogenous marmoset NTCP did not show HBV infection, in spite of supporting HBV preS1 binding by marmoset NTCP? If PMH lost high level of marmoset NTCP during cell culture and expressed only low level, how about HBV infection in PMH ectopically transduced with marmoset NTCP? And it would be nice to know the HBV infection in PMH transduced with marmoset NTCP(Q84R), NTCP(N86K/K87N), and NTCP(Q84R/N86K/K87N) given the data in Fig. 2E.
- In Fig. 5G, Infection efficiency should be shown as other figures presenting HBeAg levels. Given marmoset NTCP but not SqM NTCP expressed in HepG2 cells supported HBV infection (Fig. 5G, 2E), it would be nice to show the data on which amino acid in NTCP determines this difference in HBV-susceptibility? Again, it seems inconsistent that HBV infected in HepG2-marmoset NTCP in Fig. 5G but

HBV did not infect to PMH in Fig. 4F.

Reviewer #2 (Remarks to the Author):

Liu et al. present a functional comparison of NTCP orthologues between human, OWM and NWM. They show that marmosets could be suitable as an animal model for HBV infection and more so to a chimeric human-WM HBV. Interestingly, the authors show that while some NTCP orthologues are capable of binding PreS1 they do not support infection. Humanizing these orthologues led to HBV infection as measured by HBeAg in the supernatant of infected cells. Given the limited animal models currently available for HBV infection, developing a new infection model is needed in the field and therefore the paper is highly relevant. However, the authors should address some aspects of the manuscript prior to publication.

Major concerns:

- Throughout the manuscript, the authors utilize HBeAg, HBsAg and imaging to indicate that cells were successfully infected. To demonstrate that the different NTCP orthologues support productive HBV infection, the authors should conduct Southern blot analysis of DNA replication intermediates in infected cells to show that these cells are giving rise to all replicative HBV intermediates. Furthermore, to prove that the infection is productive, the authors should collect supernatant from infected cells and conduct secondary infection studies to show that the produced virions are indeed infectious. These studies would significantly strengthen the notion that the developed infection model in marmosets truly resembles that in human cells.
- For many of the NTCP IF images (e.g. Fig. 2D and 5G), the localization of NTCP appears to be in endosomes and not on the membrane. The authors should comment on the localization and how it compared to previously published studies. More importantly, the correlation between NTCP staining and HBc staining is poor in many IF images (e.g. Fig. 5G and suppl Fig. 7B). The authors should comment on why the correlation appears to be negative in some cells. Could this be an imaging artifact? Or is it because the NTCP is not localized to membranes in some of these cells? Since some of the NTCP staining is very intense, can the authors please show in the individual frames of each fluorophore for better visual analysis? Additionally, is the asymmetric staining found in some of the IF images (Figure 2D and 4B) common throughout the population? The authors should comment on the nature of this phenotype.
- Immunofluorescence staining of HBc is unusual and inconsistent; for example, staining in Fig. 5G and suppl Fig 7B and contradicts staining in Fig. 7A. HBc should be mostly nuclear with some diffuse cytoplasmic staining due to its nuclear localization signal. The authors should comment on why the staining is different even though these are the same cell type. Could the route of infection/transfection alter levels of HBc and the localization? The authors need to address this in the manuscript.

Minor concerns:

- It is interesting that the humanizing mutations on sqM-NTCP led to reduced PreS1 binding but higher infection rates (Suppl Fig. 5). The authors should comment on this interesting observation in the manuscript and should discuss why reduced binding may lead to enhanced infection rates.

Reviewer #3 (Remarks to the Author):

Hepatitis B virus (HBV) has a narrow host range, which limits the development of convenient animal models to study HBV infection and treatment. A major species barrier for HBV infection lies at the level of the host cell receptor for the virus, the sodium taurocholate co-transporting polypeptide

(NTCP). The authors compared NTCP from humans and other primates (Old World monkeys, New World monkeys and prosimians) and determined key residues of NTCP responsible for HBV binding and internalization. This part of the work largely confirms but also extends previous work on the same subject. In the second part, the authors tested if marmosets can serve as a suitable candidate for HBV infection as their NTCP molecules naturally harbor the residues crucial for HBV entry. They showed that primary marmoset hepatocytes and induced pluripotent stem cell-derived hepatocyte-like cells appear to support low levels of HBV, and more efficiently, woolly monkey HBV (WMHBV) infection. The authors then constructed a chimeric HBV genome harboring residues 1-48 of WMHBV preS1, considered to be the primary NTCP binding region of the viral L envelope protein, and showed that the chimeric HBV construct could infect marmoset cells more efficiently than wild-type HBV. This second part of the current work is novel and could be highly significant for the development of a more convenient primate model of HBV infection. However, a number of important technical caveats remain, which need to be addressed to strengthen the authors' conclusions.

Major issues:

1. A major issue relates to how HBV infection was measured. How did the authors exclude the possibility that the HBsAg, HBeAg, HBcAg, HBV DNA and RNA detected in the infected cells may represent, partially or completely, input signals since all these materials were present in the inoculum? This is by no means a trivial issue given the high amounts of inoculum used and the low levels of replication reached in the current cell culture systems. This issue needs to be more vigorously addressed to clearly demonstrate a productive infection, which was inefficient at best in many of the experiments the authors report here.
2. The authors showed some results measuring cccDNA, which should not be present in the inoculum, by PCR. However, PCR-based methods for cccDNA detection and quantification are well known to be complicated by the fact that the HBV rcDNA in the inoculum can be detected also albeit at lower efficiency. Since the input rcDNA and that produced in the cell following productive infection far outnumber cccDNA, PCR detection of cccDNA under these conditions have to be carefully controlled. The author did not appear to include any methods on how they measured cccDNA. They should report the PCR method details and show how their methods detected only cccDNA but not rcDNA. Alternatively, they can use Southern blot analysis, which remains the gold standard for cccDNA detection.
3. The apparently discordant results among the infection assays, e.g., HBsAg and HBeAg secretion (Fig. 5A and supplem Fig. 6D), intracellular HBcAg staining (Fig. 4G, supplem Fig. 7B and 8A), cccDNA PCR (supplem Fig. 6F), add to the concern as to whether productive infection was indeed established and whether it was accurately measured. These should be clarified with additional experiments suggested above.

Minor issues:

4. Fig. 1. More clear and consistent labels are needed to match the host species in panel A vs. B-D so that it will be clear to the reader what are the exact NTCP sequences from OWM, SqM, Mammost, and Prosimian species used in B-D?
5. Fig. 2. It is clear that residues 84-87 of NTCP, while not critical for preS1 binding, is critical for infection, a conclusion also reached by others previously. Indeed, mutants in this region enhanced infection (Fig. 2E and supplem Fig. 5B) but decreased preS1 binding (supplem Fig. 5A). Based on these results and the recently published structural data of NTCP and NTCP-preS1 complex, can the authors provide an plausible explanation(s) for these results.
6. Fig. 4C. "HBsAg" should be labeled as WMHBsAg to avoid confusion.

Reviewer #4 (Remarks to the Author):

This carefully designed study describes the generation of the simian tropic virus and the establishment of a marmoset animal model for HBV infection. There is a large need to develop a smaller NHP model

of HBV infection since current murine models have limitations. By conducting a mutational analysis of the HBV receptor (sodium taurocholate co-transporting polypeptide, NTCP) orthologs from different primate species, the authors determined several NTCP residues critical for HBV entry into the host cell. Additionally, the authors established primary and stem-cell-derived marmoset hepatocyte cultures that supported infection of HBV or chimeric HBV harboring 48 amino acid residues from woolly monkey HBV. The study is well-designed, the manuscript is clearly written, and the results constitute a significant research accomplishment in the HBV field. However, structures from panel 3C need to be carefully inspected before the manuscript is accepted for publication.

Major comments:

1. It is not clear why the mutagenesis analysis in the 84-87 regions was not performed for OWM NTCPs. In particular, it would be interesting to see the effect of mutations in residues 84 and 86. Additionally, have the authors performed mutagenesis experiments for marmoset NTCPs?
2. In figure 3C, It appears as hNTCP and SqMWT structure alignments might have been switched. Amino acids labels look correct, but the amino acid side chains do not correspond with the amino acid labels.
3. Again, in figure 3C, SqMK87N does not appear to have a side chain corresponding to asparagine at position 87. Please check all models from panel 3C.

Minor comments:

1. Line 65, "mic" should be changed to "mice"
2. Throughout the text, "cyro-EM" should be changed to "cryo-EM" (lines 122, 861 in the main text; lines 88, 93 in the supplemental figures file)
3. Line 307, Can you please clarify whether H₉₆relhuman-96 SACC-PMHs were used in this study? From the experiment setup, it appears as SACC-PMHs were constructed from marmoset, not human hepatocytes
4. Line 850, Please add "pre S1" before "binding assay"
5. Lines 863-868, panels C and D need separate descriptions.
6. Line 872, Figure 2E needs to have description for ****
7. Lines 874-881, please specify whether any PDB models were used for structure alignment
8. Figure 1D, X-axis labels (6 and 8), need to be rotated 90° clockwise
9. In figure 3C, please remove hydrogen atoms from all models

Reviewer #1 (Remarks to the Author):

The authors examined the ability of NTCP derived from OWM, NWM, and prosimian as well as human for supporting HBV preS1 binding and infection. They further examined the mutant NTCPs to show that HBV infection was reproduced by R158G and further promoted by R158G/P165L in OWM-derived NTCP, and was observed by either Q84R, K87N, or additional combined mutations in SqM-NTCP. Structural prediction suggested that the mutation R158G/P165L in OWM NTCP changed the stability of the loop between TM4-5, which might be related with the elevated HBV infection. They showed the infection of WMHBV in primary marmoset hepatocytes. The HBV/WMHBVpreS1(1-48), a chimeric virus of which preS1(2-48) of HBV was replaced by that of WMHBV, was shown to infect HepG2-marmoset NTCP cells, primary marmoset hepatocytes, and iPS-derived hepatocyte-like cells more efficiently than HBV. These information is important for establishing an in vivo surrogate infection model of HBV. The authors showed a lot of data for infection susceptibility of NTCP derived from different species and their mutants, which helps to understand the regulation mechanism of hepadnavirus entry. However, there are multiple inconsistent results between figures or those not well explained. The conclusion of this study is not clear. If the authors want to claim that the HBV/WMHBV preS1 chimera is useful for in vivo hepadnavirus infection model, they should show in vivo infection of this chimeric virus to marmosets.

If the authors focused on the amino acids in NWM- or OWM-NTCP that determine the HBV susceptibility, majority of the results are already reported, including the importance of aa 158 of NTCP. The story of the paper has to be totally reconstructed to claim the significance of the priority of this paper. Other comments are shown below.

Author response: We thank this reviewer for their assessment of our study and highlighting the significance of our work for understanding HBV species tropisms. As this reviewer points out, several prior studies – all of which were cited in our manuscript – aimed at delineating which differences in amino acid sequences of different NTCP orthologues affect HBV entry (Müller *et al.*, *PloS One*, 2018, PMID: 29912972; Yan *et al.*, *eLife*, 2012, PMID: 23150796; Jacquet *et al.*, *JVI*, 2019, PMID: 30541833; Ni *et al.*, *Gastroenterology*, 2014, PMID: 24361467; Takeuchi *et al.*, *JVI*, 2019, PMID: 30541857). To our knowledge, these papers reported on the differences in human NTCP, and New World monkey or Old World monkey orthologues, respectively, and pointed out that a single R158R adaptive mutation in Old World monkey NTCP could confer HBV susceptibility. However, in contrast to our present study, these papers did not further identify other key residues that are responsible for HBV binding and internalization, respectively. In particular, for New World monkey NTCP orthologues, the aa 84-87 region has not previously been thoroughly investigated.

In this study, through extensive step-by-step mutational analysis and screening of NTCP orthologues from Old World monkeys, New World monkeys and prosimians, we functionally characterized NTCP residues and determined key residues responsible for viral binding (glycine 158, Leucine 165) and internalization (aa 84-87), respectively. We also determined – to our knowledge for the first time -, that woolly monkey HBV (WMHBV) can efficiently infect hepatocytes from a New World monkey species - marmosets. Strikingly, we found that marmoset NTCP naturally harbors an arginine residue (the same as human) in position 84, which is crucial for HBV entry and can support some level of HBV infection. Furthermore, we went one step further and generated a simian-tropic HBV by targeted viral adaptation (replacing the small part of HBV preS1 with WMHBV preS1) and successfully increased HBV infectivity on marmoset hepatocytes (??), which holds promise to be used for modeling HBV infections *in vivo*. Thus, we believe that our work represents considerable novelty. We have included numerous new pieces of evidence in our revised manuscript to further substantiate our claims, but kept the overall flow of the story which we believe conveys clearly the mainly points we intend to relay.

- In Fig. 2, the authors showed that SqM NTCP as well as OWM NTCP(R158G) supported HBV preS1 binding. Why SqM NTCP (WT) did not render HBV infection while OWM NTCP(R158G) supported a low level of HBV infection?

Author response: We thank this reviewer for raising this point which we believe is one of the important findings of this study. NTCP sequence alignment and phylogenetic analysis of NTCP orthologues of diverse species validated that two regions - aa 84-87 and aa 157-165 - are different between human and Old World monkey (OWM) or New World monkey (NWM) (**Fig. 1A**). Differences in these two regions result in incompatibilities between HBV and simian NTCP orthologues and ultimately, failure of HBV infection.

For OWM NTCPs, the main differences lay in aa 157-165 region when comparing it to human NTCP (**Fig. 1A**) which is substantiated by our data and prior studies (e.g. Müller *et al.*, *PLoS One*, 2018, PMID: 29912972; Yan *et al.*, *eLife*, 2012, PMID: 23150796; Jacquet *et al.*, *JVI*, 2019, PMID: 30541833; Takeuchi *et al.*, *JVI*, 2019, PMID: 30541857). R158G is the critical for HBV infection as evidenced by the fact that wild type OWM NTCPs harboring an arginine in position 158 neither support HBV binding, nor infection (**Fig. 1C&1D**). The reason why OWM NTCP (R158G) only supports HBV infection inefficiently is that in addition to residue 158, other residues in the aa 157-165 region are also important contributors to HBV uptake. In this study, we also determined that amino acid 165 plays a key role. Indeed, humanizing both R158G and P165L in OWM NTCP significantly increased HBV binding and infectivity, considerably more so than the R158G single mutation (**Figure 2E, Supplementary 5A, 5B & 5E**). We also went one step further and deciphered the underlying mechanisms using *de novo* derived structural models. We show that P165 in OWM-NTCP vs L165 in hNTCP may disrupt the formation of a helix that stabilizes a loop structure between TM4 and TM5 (**Fig. 3B**) and helps control the distance and conformation between the core and panel domains to maintain the function of the tunnel and facilitate HBV infection.

To gain insights to the barriers of HBV transmission in NWM, we focused exemplarily on SqM. We determined that wild type SqM NTCP supports HBV binding, but SqM NTCP does not support HBV virion entry/internalization (**Fig. 2C, 2D, 2E**), which points to the fact that differences in the aa 84-87 region in NWM preclude efficient HBV entry. We analyzed systematically the effects of single and combinations of mutations in positions Q84, N86 and K87 and found that humanizing individually Q84R, and K87N increases the ability of SqM NTCP to support HBV infection. This increase is even greater when both of these mutations are present. (**Fig. 2E**).

Also, substitutions Q84R, K87N, N86K/K87N, and Q84R/N86K/K87N in SqM NTCP presented HBV infection. Do these mutations facilitate the HBV preS1 binding compared with the wild type SqM NTCP? The authors should present data to make the relationship between the HBV preS1 binding and HBV infection clearer.

Author response:

We thank this reviewer for raising this interesting point. For the substitutions Q84R, K87N, N86K/K87N, and Q84R/N86K/K87N in SqM NTCP, we had already performed binding assays and our results showed that substituting Q84R slightly reduced the HBV preS1 peptide binding, while for substitutions of K87N, N86K/K87N, and Q84R/N86K/K87N in SqM NTCP, preS1 peptide binding decreased more significantly as compared to the wild type SqM NTCP (please see **Supplementary Figure 5A**). To further validate the findings and following this reviewer's suggestion, we also introduced humanizing mutations in marmoset NTCP as another NWM representative. Since marmoset NTCP naturally harbors an arginine in position 84, we only introduced K87N and measured preS1 peptide binding by flow cytometry. Consistent with the

SqM NTCP data, substituting K87N in marmoset NTCP also decreased HBV preS1 peptide binding significantly as compared to wild type marmoset NTCP (**Supplementary Fig. 5C&D**).

HBV uptake is a complex process which depends on initial attachment of the virion to heparan sulfate proteoglycans and subsequent high-affinity interactions of the myristoylated preS1 domain with NTCP (Schulze et al. *Hepatology*, 2007; PMID: 18046710; Liang et al. *Hepatology*, 2015; PMID: 26239691, Yan et al. *eLife*, 2012) which is thought to trigger receptor-mediated endocytosis. (Huang et al. *J Virol*, 2012; PMID: 22740403; Perez-Vargas et al. *eLife*, 2021; PMID: 34190687). Despite considerable advances in delineating the initial stages of this process, later steps in the uptake of the virion, i.e. viral and endosomal fusion, release of the nucleocapsid from the endosome, are less well defined. Conceivably, HBV depends on additional entry factors as suggested by some (Liang et al. *Hepatology*, 2015; PMID: 26239691) and conformational changes in the NTCP triggered by virion binding might be needed for endocytosis.

In this study, we took advantage of the fact that NTCP orthologues from different primate species show different abilities to support HBV binding and/or infection. Comprehensive analysis of the residues in crucial domains allowed us to delineate those amino acids in NTCP that are needed for binding vs infection. Building on prior data, we highlighted the importance of additional amino acids within the aa157-165 motif, which is part of the transmembrane 5 forming the functional tunnel (Fig. 2A&Fig.3A). Consistent with prior data, R158G enables binding of HBV to OWM-NTCP (gain-of-function) while G158R abrogates HBV binding to SqM-NTCP (loss-of-function) (**Fig. 2C, D; Supplementary Fig. 4B, C**), underscoring that G158 is necessary for HBV binding. Combining structural analysis using both previously published cryoEM data and *de novo* generated models of NTCP mutants and extensive functional (binding and infection) data, has allowed us to identify L165 as an additional important residue modulating HBV binding and infection (**Fig. 2E; Supplementary Fig. 5A&5B&5E**).

Interestingly, SqM NTCP supports preS1 peptide binding but it does not support infection. (**Supplementary Fig. 5A&5B&5E; Fig. 2E**). This led us to investigate the role of the aa84-87 motif in more detail, which is part of a loop domain between transmembrane 2-3 (**Fig. 2A&B**). Our data are consistent with the notion that the region is responsible for the extracellular interaction between HBV and NTCP, subsequently inducing topological/conformational changes to initiate viral entry (**Fig. 2E**). Another piece of evidence suggesting that the aa84-87 motif is responsible for HBV entry but not binding, is humanizing mutants, including Q84R, K87N, N86K/K87N, and Q84R/N86K/K87N in SqM NTCP, which we have identified facilitates HBV infection (**Fig. 2E**) but decrease HBV binding (**Supplementary Fig. 5A**).

Altogether, we present data delineating the distinct functions of different NTCP motifs necessary for binding and infection which provide new insights into the complex interactions between NTCP and the virus.

- In Fig. 3, the authors speculated that the substitution at aa 165 in NTCP affect the structure of the TM4-5 region. The authors should at first mention the possibility that P165L in OWM NTCP may directly augment the binding of this site to HBV preS1 and examine this possibility.

Author response:

We thank this reviewer for raising this point which is one of our important findings in this study. Data from our structural analysis (**Fig. 3**) suggest that P165 in OWM-NTCP vs L165 in hNTCP may disrupt the formation of a helix that stabilizes a loop structure between TM4 and TM5. Accordingly, we introduced an P165L mutation in OWM NTCP and tested the ability of this mutant to support HBV preS1 binding and HBV infection. In the updated **Supplementary Fig. 5A&5B&5E**, we now demonstrate that the R158G&P165L double mutation significantly enhances

HBV preS1 binding and HBV infection as compared to the R158G single mutation in OWM NTCP (Fig. 2E).

And then about the possibility the authors proposed in Fig. 3B, they should introduce mutations in the TM4-5 region to disrupt the structure seen for OWM (R158G/P165L) to examine whether the proposed hypothesis is true or not.

Author response:

We thank this reviewer for this comment. In this study, we determined that the humanizing P165L mutation in TM5 of OWM NTCP can form a helix, the same as human NTCP, between the loop region of TM4 and TM5 that appears to stabilize the loop structure between TM4 and TM5. We have human NTCP acting as the positive control and wild type OWM NTCP and R158G OWM NTCP acting as the negative controls (Fig. 3B). On the other hand, the protein structure is formed based on the biophysical properties of the amino acid sequences and the interactions among surrounding amino acids including but not limit to van der Waals force, net charge, isoelectric point, avg. hydrophathy, aliphatic index and instability index. Here, the helix formation inside of the TM4-5 loop is affected by the residues in TM5 (not the residues within the loop since they are the same among human NTCP), wild type OWM NTCP, R158G OWM NTCP and R158G&P165L OWM NTCP (Fig. 3B). P165L mutation in TM5 mainly changed the instability index of this TM4-5 region so a more stable helix structure is formed (Fig. 3B).

- In Fig. 3C, the reviewer doesn't understand how SqM NTCP(K87N) and marmoset NTCP are structurally more similar to human NTCP. The authors should show a quantitative index.

Author response:

We apologize for not having presented the data more clearly in our initial submission. Our data show that SqM NTCP^{K87N} and marmoset NTCP are indeed more similar to human NTCP with respect to the side chains in this loop region. To make this point clearer we re-analyzed the NTCP structural data. To highlight the important residues (position 84 and position 87) which are responsible for HBV infection that we found in this study, we highlight the sidechains in red in a stick model. Furthermore, we have now also included quantitative indices. Hydrophobicity as the grand average of hydrophobicity (GRAVY) to represent the hydrophobicity value, and the Instability Index as a measure for the stability of the 84-87 peptide is shown accordingly. We also analyzed the structure similarity between hNTCP and SqMNTCP and its mutants and plotted the alignment distances (shown as Å) for each amino acid. The revised data is shown in the updated Fig. 3C and Supplementary Fig. 3C.

- Fig. 3D calculated RMSD of NTCPs. Since only a part of regions in NTCP, not the entire region, is involved in preS1 binding, it would be nice to present the similar analysis using the 84-87 and 157-165 aa and the surrounding regions, instead of using the entire NTCP.

Author response:

Following the reviewer's suggestion, we expanded our structural alignment analyses for the 84-87 region and the surrounding regions. Since 84-87 region localizes to a loop between the transmembrane (TM) 2 and TM 3(a) of the NTCP core domain, the structures of 84-87 region plus surrounding TM2 and TM3 among different NTCP orthologs were shown separately for comparison (Supplementary Fig. 3D). However, since the overall structures are very similar for this region and the alignment distance (shown as RMSD score) of all orthologues to human NTCP is around 0.1 Å, we would like to remain cautious not to overinterpret the structure similarities based on the very small RMSD scores. The important differences are localized within the side chains of 84-87 amino acids which may affect the ability of the receptor to interact with HBV and/or can conceivably influence conformational changes needed to promote viral entry. The revised data are shown in the updated Figure 3C and Supplementary Fig. 3D. The 157-165 region at

the N-terminal end of TM5 in the panel domain is indeed of great importance for HBV preS1 binding because it is part of the functional tunnel and binding pocket for bile acid as well as HBV preS1.

The reviewer is entirely correct that only a part of the regions within NTCP are responsible for HBV binding. However, since the functional cavity is composed of multiple transmembrane domains - specifically TM1, 3, 5, 6, 8 and 9 - which already cover most of the entire NTCP molecule (TM1-9), it would be difficult to pull specific amino acids out of this conformational context to perform a separate structural alignment. Furthermore, NTCP is a transmembrane transporter and there are “active open” and “inactive closed” conformations that switch dynamically. These conformational changes depend on different TM and amino acids in different regions to work in concert (Goutam et al. Nature, 2022; PMID: 35545671; Asami et al. Nature, 2022; PMID: 35580629). For these reasons, we also show the whole NTCP structure alignment.

- Fig. 4B should show images displaying several cells, not a single cell, to avoid unexpected bias when selecting the images.

Author response:

Following the reviewer’s suggestion, the experiments were repeated and images with wider fields were used to show the binding (see **new figure 4B**). To further support the imaging data, additional flow based binding experiments were performed, which allowed us to gate on NTCP expressing cells (RFP channel) to quantify HBV preS1 binding cells (FITC channel) (**Supplementary Fig.2C**). In the flow cytometry assays, at least 3,000 NTCP expressing cells were captured and analyzed and at least three independent experiments were performed. The results are summarized in the new figures **Fig. 4A**, **Supplementary Fig. 4C**, **Supplementary Fig. 5A**, and **Supplementary Fig. 7A**.

- In Fig. 4F, why non-transfected PMH producing endogenous marmoset NTCP did not show HBV infection, in spite of supporting HBV preS1 binding by marmoset NTCP? If PMH lost high level of marmoset NTCP during cell culture and expressed only low level, how about HBV infection in PMH ectopically transduced with marmoset NTCP? And it would be nice to know the HBV infection in PMH transduced with marmoset NTCP(Q84R), NTCP(N86K/K87N), and NTCP(Q84R/N86K/K87N) given the data in Fig. 2E.

Author response:

We apologize for the unclear labeling which may have led to the misinterpretation of our results. In Fig. 4F, the control group labeled with “PLVX-mock” was the group in which the PMHs were transduced with “PLVX vehicle” empty lentiviruses and without HBV inoculation (mock). Since no HBV was added, there is no infection in this group. The experimental group labeled with “PLVX-hNTCP” was the group in which the PMHs were transduced with “PLVX-hNTCP” lentiviruses to express hNTCP and then subsequently infected with HBV. To label these two groups more accurately, “PLVX-mock” was revised to be “PLVX-vehicle-mock” and the “PLVX-hNTCP” was revised to be “PLVX-hNTCP-HBV” in the revised **Fig. 4F**.

Our rationale for the experiment leading to the data shown in Fig. 4E, 4F and 4G is based on our observation that marmoset NTCP supports WMHBV preS1 binding (**Fig.4A&4B**) and infection of HepG2-marmoset NTCP expressing cells (**Fig.4C**). Thus, we attempted WMHBV infection of primary marmoset hepatocytes (PMHs) —arguably the most physiologically relevant cell type for HBV/WMHBV in vitro infection taking advantage of our previously established (Winer et al. Hepatology, 2020, PMID: 31206195) self-assembling co-cultures (SACCs) of PMH and murine non-parenchymal stromal cells (**Fig. 4D**), which can maintain the highly differentiated phenotype of primary hepatocytes for at least 4 weeks (**Supplementary Fig. 7B**). We aimed to determine the natural infection, and thus did not include conditions in which marmoset NTCP (or its mutants)

was ectopically expressed. Since some species (e.g. mouse, rat) do not support HBV/WMHBV whole life cycle, we investigated whether PMHs support HBV replication by transducing 1.3x HBV replication competent genome (**Fig.4E**) and whether PMHs support HBV life cycle after the entry process by overexpressing hNTCP and HBV infection (**Fig.4F**). Finally, we provide evidence that PMH can indeed support WMHBV infection robustly and all post entry steps of HBV. These results provided a clear rationale for constructing HBV/WMHBV chimeric virus that we characterized in detail (please see **Fig. 5**).

- In Fig. 5G, Infection efficiency should be shown as other figures presenting HBeAg levels. Given marmoset NTCP but not SqM NTCP expressed in HepG2 cells supported HBV infection (Fig. 5G, 2E), it would be nice to show the data on which amino acid in NTCP determines this difference in HBV-susceptibility?

Author response:

We thank this reviewer for these comments. Based on our transfection data on HepG2 cells, we compared the antigen expression and viral replication abilities of WMHBV, HBV and the HBV/WMHBV preS1 chimeric viruses. We found that HBeAg levels are significantly higher (>3 folds, $p < 0.0001$) following transfections with WMHBV than those with HBV or HBV/WMHBV preS1 chimeric viruses. Nonetheless, the HBsAg and HBV DNA levels are similar (**Supplementary Fig.7D**). Conceivably, the basal core promoter and core promoter activity (which drives HBeAg expression) of WMHBV is higher than that of HBV in HepG2 cells. On the other hand, the infection efficiency of marmoset NTCP expressing HepG2 cells with HBV is much lower as compared to HBV infections of hNTCP expressing HepG2 cells,. We believe that this might explain why we did not detect significant levels of HBeAg in marmoset NTCP expressed HepG2 system and thus only showed the HBsAg levels (**Fig.5G**). We expanded the discussion to adequately reflect these points (lines 274-283).

To provide additional evidence for productive infections, we also carried out Southern blots on infected marmoset NTCP expressing HepG2 cells. Following infection with HBV or our HBV/WMHBV preS1 chimeric virus, we detected bands corresponding to HBV DNA replication intermediates (updated **Fig.5I**). In addition, we also quantified HBV cccDNA by qPCR using stringent protocols and conditions, including HIRT DNA extraction, T5 digestion and a cccDNA specific qPCR program (Allweiss et al. *Gut*, 2023; PMID: 36707234; Xia et al. *Methods in Molecular Biology*, 2017; PMID: 27975308) Using this cccDNA qPCR approach, we were able to detect cccDNA both in HBV and our HBV/WMHBV preS1[1-48] chimeric virus infection groups. Importantly, the cccDNA copy numbers were higher in the HBV/WMHBV preS1[1-48] chimeric virus than in the HBV infection group (updated **Fig. 5H**). We were also able to detect HBV RNA by in situ hybridization (ISH) and HBc by immunohistochemistry (IHC) (updated **Fig. 5J**).

We undertook considerable efforts to determine the genetic determinants within NTCP from diverse primate lineages affecting HBV infection. By comparing NWM and OWM NTCP, we found that OWM NTCP neither supports HBV binding, nor infection because of the R158 and P165 residues. We noticed that NWM NTCP (SqM) supports HBV binding, but not viral entry. This prompted us to investigate systematically the underlying cause which led us to the differences in amino acids 84-87 within NWM NTCP which preclude entry. By introducing the humanizing Q84R mutation or even better, the K87N mutation on SqM NTCP, we found these single mutations can convert HBV infectivity (**Fig. 2E**). More importantly, we found compared with SqM NTCP, marmoset NTCP naturally harbor R84, which is same as hNTCP. So, given these data, we believe the amino acid in position 84 (Q for SqM NTCP, R for marmoset NTCP, **Fig.1A**) is a determinant for the difference in HBV-susceptibility for SqM and marmoset. However, since SqM and marmoset NTCP have different amino acids in other regions, except aa 84-87 and aa 157-165

regions, future studies would be required to portray whether these regions also play some roles that may affect HBV infectivity.

Again, it seems inconsistent that HBV infected in HepG2-marmoset NTCP in Fig. 5G but HBV did not infect to PMH in Fig. 4F.

Author response:

The comments were addressed. Please see our previous answers regarding this point.

Reviewer #2 (Remarks to the Author):

Liu et al. present a functional comparison of NTCP orthologues between human, OWM and NWM. They show that marmosets could be suitable as an animal model for HBV infection and more so to a chimeric human-WM HBV. Interestingly, the authors show that while some NTCP orthologues are capable of binding PreS1 they do not support infection. Humanizing these orthologues led to HBV infection as measured by HBeAg in the supernatant of infected cells. Given the limited animal models currently available for HBV infection, developing a new infection model is needed in the field and therefore the paper is highly relevant. However, the authors should address some aspects of the manuscript prior to publication.

Author response:

We thank this reviewer for pointing out the importance of our study for the development of new animal models for HBV infection.

Major concerns:

- Throughout the manuscript, the authors utilize HBeAg, HBsAg and imaging to indicate that cells were successfully infected. To demonstrate that the different NTCP orthologues support productive HBV infection, the authors should conduct Southern blot analysis of DNA replication intermediates in infected cells to show that these cells are giving rise to all replicative HBV intermediates. Furthermore, to prove that the infection is productive, the authors should collect supernatant from infected cells and conduct secondary infection studies to show that the produced virions are indeed infectious. These studies would significantly strengthen the notion that the developed infection model in marmosets truly resembles that in human cells.

Author response:

This reviewer makes an important point. Following their suggestion, we performed Southern blot analysis of HBV DNA in the marmoset NTCP expressing HepG2 cells infected with HBV or our HBV/WMHBV preS1[1-48] chimeric virus. Our results show bands corresponding to different HBV DNA replication intermediates (rcDNA, dsDNA, ssDNA and other smeared signals indicating different lengths of negative single strands or double strand DNA with variety length of HBV positive strand) (please see the revised **Fig. 5I**). More importantly, and consistent with our other infection data in Fig. 6, the Southern blot signal is stronger in cells infected with the HBV/WMHBV preS1[1-48] chimeric virus than those infected with HBV.

In addition to Southern blotting, we also carried out qPCR assays to quantify cccDNA following the most updated HBV cccDNA qPCR quantification protocol (Allweiss et al. *Gut*, 2023; PMID: 36707234; Xia et al. *Methods in Molecular Biology*, 2017; PMID: 27975308) by Hirt DNA extraction, T5 digestion and cccDNA specific qPCR program. Our cccDNA qPCR further ascertain the presence of cccDNA both in HBV and our HBV/WMHBV preS1[1-48] chimeric virus infection groups. Notably, the cccDNA levels in the HBV/WMHBV preS1[1-48] chimeric virus infection group is higher than that in the HBV infection group (revised **Fig. 5H**).

To provide additional evidence for the infection we also carried out HBV RNA in situ hybridization (ISH) and HBc immunohistochemistry (IHC) on Formalin Fixed Paraffin Embedded (FFPE) samples. Using previously validated, commercially available, probes we detected HBV RNA in infected cells (**Fig. 5J**). Collectively, all parameters measured using multiple orthogonal methods, HBsAg (by ELISA, cccDNA (by qPCR), HBV DNA (by Southern blotting), HBV RNA (by ISH) and HBc (by IHC) are consistent with *de novo* infection.

We appreciate the reviewer's suggestion for the supernatant reinfection experiments. However, for HBV infection, this is probably not possible. To our knowledge no robust infections have been established in the HBV field attempting to propagate virus from cells infected *in vitro*. As this reviewer is aware, supernatants from infected cells yield only low viral titers and usually very high HBV doses are needed to establish infections. Thus, in the field, HBV virus stocks for infections

are generally derived from three routes, none of which are suitable for carrying out the experiment suggested by the reviewer:

1. HBV transgenic cell lines, including but not limited to the widely used HepAD38 (Ladner et al. *Antimicrobial Agents and Chemotherapy*, 1997; PMID: 9257747) and HepG2.2.15 (Sells et al. *PNAS*, 1987; PMID: 3029758) and other recently generated HBV-expressing cell lines (e.g. Zhang et al. *J Hepatol*, 2021; PMID: 34363922). These producer cells usually have to be cultured at large scale to harvest enough supernatants for virus concentration and purification.
2. Hepatoma cell lines transfection with HBV replication competent clones to produce high HBV titer supernatants (e.g. Liu et al. *JHEP Reports*, 2022; PMID: 36035359). Even so, usually the supernatants still need to be concentrated to harvest virus stock with good quality for infection.
3. High HBV viremia patient serum or human liver chimeric mouse serum (e.g. Liu et al. *Hepatology*, 2020; PMID: 31278760; Liu et al. *JHEP Reports*, 2022; PMID: 36035359). These serum samples usually can be used for HBV infection directly because of the high HBV titer and virion inactivity.

- For many of the NTCP IF images (e.g. Fig. 2D and 5G), the localization of NTCP appears to be in endosomes and not on the membrane. The authors should comment on the localization and how it compared to previously published studies.

Author response:

We thank this reviewer for raising this point. We think that the seemingly limited NTCP localization on the plasma membrane can be attributed to the overexpression of fluorescently tagged NTCP, which can result in considerable background signal. To address this issue, we repeated the overexpression experiments with hNTCP, OWM-NTCP, SqM-NTCP, marmoset-NTCP and prosimian-NTCP and optimized our imaging settings which allowed us to confirm membrane localization for all five NTCP orthologues. These new imaging data are shown in the updated **Supplementary Fig. 2B**. We also repeated the HBV binding experiments in Fig. 2D and the NTCP expression in both wild type and mutants' groups are well localized on the cell membranes (updated **Fig. 2D**).

Following the reviewer's suggestion, we also reviewed other studies in which NTCP had been expressed ectopically. For convenience we took the liberty to reproduce figure 1B of Li et al. (*Gastroenterology* in 2014 (PMID: 24361467)) below, demonstrating that also here NTCP signal was not strictly localized to the cell membranes.

[redacted]

Fig. 1B (Ni et al. *Gastroenterology* 2014; PMID: 24361467)

More importantly, the correlation between NTCP staining and HBc staining is poor in many IF images (e.g. Fig. 5G and suppl Fig. 7B). The authors should comment on why the correlation appears to be negative in some cells. Could this be an imaging artifact? Or is it because the NTCP is not localized to membranes in some of these cells? Since some of the NTCP staining is very intense, can the authors please show in the individual frames of each fluorophore for better visual analysis? Additionally, is the asymmetric staining found in some of the IF images (Figure 2D and 4B) common throughout the population? The authors should comment on the nature of this phenotype.

Author response:

We thank this reviewer for their suggestions. We repeated the imaging data, and the individual fluorescence channels were shown in **Supplementary Fig.9**. Our data show that NTCP expression is reduced in HBc+positive cells. To confirm this observation, we also performed multiplexed HBV RNA in situ hybridization (ISH) and HBc immunohistochemistry (IHC) by using Formalin Fixed Paraffin Embedded (FFPE) marmoset-NTCP expressing HepG2 cells. Consistent with our HBc immunofluorescence data, cells harboring HBV RNA or HBc had decreased NTCP signals (**Fig. 5J**). This phenomenon can possibly be explained by the reported receptor internalization during HBV infection (Liang et al. Hepatology, 2015; PMID: 26239691) after which NTCP may be subject to degradation. Another possible and not necessary mutually exclusive explanation could be superinfection exclusion, the ability of established virus infection to interfere with secondary virus infection by causing downregulation of critical entry factors, which has been documented for numerous other viruses including but not limited to Newcastle disease virus (NDV) (Bratt et al. Virology, 1968; PMID:5662868), Rous sarcoma virus (RSV) (Steck et al. Virology, 1966; PMID: 4287699), bovine viral diarrhea virus (BVDV) (Lee et al. J Virol, 2005; PMID: 15731218) and hepatitis C virus (HCV) (Tscherne et al. J Virol, 2007; PMID: 17287280). This is certainly an interesting phenomenon whose mechanistic delineation is not within in the scope of the present study and will need to be investigated in the future.

We also repeated the NTCP expression experiments without HBV infection and optimized the imaging settings and our results confirmed that the NTCP is localized to the membranes (**Supplementary Fig. 2B**). The images in the original Fig. 2D and 4B represent HBV preS1 binding data. We repeated these experiments and took pictures that include more cells per field. Our updated results (updated **Fig. 2D** and **Fig. 4B**) show that most of the NTCP signal (red) can be found on the plasma membrane where it colocalizes with the HBV preS1 signal (green, overlap of the signal in the merged images is shown in yellow). Nonetheless, there is some heterogeneity with respect to membrane localization which is consistent with prior studies (e.g. Ni et al. Gastroenterology, 2014; PMID: 24361467), and was discussed in our response to the previous point.

- Immunofluorescence staining of HBc is unusual and inconsistent; for example, staining in Fig. 5G and suppl Fig 7B and contradicts staining in Fig. 7A. HBc should be mostly nuclear with some diffuse cytoplasmic staining due to its nuclear localization signal. The authors should comment on why the staining is different even though these are the same cell type. Could the route of infection/transfection alter levels of HBc and the localization? The authors need to address this in the manuscript.

Author response:

We thank this reviewer for raising this point. We repeated the experiments and reanalyzed the HBc localization both following transfection of the HBV genome and after HBV infection. First, after transfecting a plasmid encoding an HBV 1.3x infectious clone into HepG2 cells we found that the HBc signal is either in the cytoplasm near the nucleus or localizes within the nucleus (**Supplementary Fig. 8A**). To confirm the cytoplasmic staining, we also used residual lab stocks of the Dako anti-HBc antibody (B0586, Dako, Denmark) which has been widely used for detecting

HBc by immunofluorescence (e.g. in Schulze et al. *Hepatology* 2011; PMID: 21953491), but this particular antibody is not commercially available anymore. Our data show that after transfection, HBcAg mostly displayed a cytoplasmic distribution with some signals in the nucleus (**Supplementary Fig. 8B**). Five days after infection, HBcAg also predominantly localized in the perinuclear areas with some signals in the nuclear.

The reviewer points out correctly that there are at least two potential nuclear localization signals (NLS) in arginine-rich carboxyl terminus (Eckhardt et al. *J Virol*, 1991; PMID: 1987370; Kann et al. *J Cell Biol*, 1999; PMID: 10189367) but their exact location remains to be resolved. However, it is well established that HBc antigen traffics within hepatocytes to carry out different functions. HBV capsids are synthesized in the cytoplasm and indeed, Li et al. identified two nuclear export signals or cytoplasmic retention signals for the core protein to exit the nucleus in the arginine rich domain (ARD) at the C-terminus of HBc 147–183 where four stretches of clustering arginine are contained (Li et al. *PLoS Pathog*, 2010; PMID: 21060813). The nuclear export signals act as the shuttling signal for the HBc trafficking between nucleus and cytoplasm.

We also compared the HBcAg localization with the papers published previously and particularly Khakpoor et al. found that HBcAg displayed a predominantly cytoplasmic distribution during early phases of infection (before days 14) and the HBcAg antigen would go to the nucleus after prolonged HBV infection (after days 21) (Khakpoor et al. *J Virol.*, 2019; PMID: 30518652).

[redacted]

(Khakpoor et al. *J Virol.*, 2019; PMID: 30518652)

Data from a very recent paper also suggest that HBV core antigen would translocate from the nucleus to the cytoplasm during cell division (Romero et al. *mBio*, 2023; PMID: 36809075). Clinically, hepatic HBc is mainly distributed in the nuclei in patients with mild hepatitis while HBc appears to predominantly localize to the cytoplasm in patients with chronic active liver diseases (Chu et al. *Gastroenterology*, 1987; PMID: 3536652; Chu et al. *Gastroenterology*, 1995; PMID: 7498658; Hsu et al. *J Hepatol*, 1987; PMID: 3655309; Naoumov et al. *J Hepatol*, 1993; PMID: 8301053).

Taken together, HBc localization is not exclusively nuclear but – as the reviewer is aware - is dynamic and differs on timing and HBV replication status. In this study, we detected the HBc by immunofluorescence after 5 day infection in vitro in the HepG2-NTCP cells, so the localization may be different with other studies in vivo or prolonged HBV infection. Nonetheless, our data are consistent with published literature.

Minor concerns:

It is interesting that the humanizing mutations on SqM-NTCP led to reduced PreS1 binding but higher infection rates (Suppl Fig. 5). The authors should comment on this interesting observation in the manuscript and should discuss why reduced binding may lead to enhanced infection rates.

Author response:

In this study, we indeed found humanizing mutations in the SqM-NTCP 84-87 region that reduced preS1 binding but enhanced HBV infection. AAs 84-87 localize in an extracellular loop between TM2 and TM3, roughly 30 Å from residue 157. None of these residues in this loop region appears to affect bile acid uptake (Yan et al. J Virol, 2014; PMID: 24390325). On the other hand, HBV binding and bile acid uptake may directly compete as HBV preS1 and bile acids share the same functional tunnel (Asami et al. Nature, 2022; PMID: 35580629). Thus, we think that the 84-87 region is important for HBV binding but plays a role in a step following HBV internalization, conceivably, in the conformational change in the NTCP topology induced by the interaction with HBV and subsequently, the initiation of endocytosis. We performed a more careful structural analysis and found that humanizing mutations in SqM-NTCP 84-87 can increase the stability of the loop and causing the hydrophobicity to be more similar to the counterpart of hNTCP (updated **Fig. 3C**). Putatively, a more rigid NTCP structure in the TM2-3 loop may not necessarily be beneficial for preS1 binding, but might help during viral entry. We have discussed these points in the revised manuscript (lines 143-145 & 250-255) as suggested by the reviewer.

Reviewer #3 (Remarks to the Author):

Hepatitis B virus (HBV) has a narrow host range, which limits the development of convenient animal models to study HBV infection and treatment. A major species barrier for HBV infection lies at the level of the host cell receptor for the virus, the sodium taurocholate co-transporting polypeptide (NTCP). The authors compared NTCP from humans and other primates (Old World monkeys, New World monkeys and prosimians) and determined key residues of NTCP responsible for HBV binding and internalization. This part of the work largely confirms but also extends previous work on the same subject. In the second part, the authors tested if marmosets can serve as a suitable candidate for HBV infection as their NTCP molecules naturally harbor the residues crucial for HBV entry. They showed that primary marmoset hepatocytes and induced pluripotent stem cell-derived hepatocyte-like cells appear to support low levels of HBV, and more efficiently, woolly monkey HBV (WMHBV) infection. The authors then constructed a chimeric HBV genome harboring residues 1-48 of WMHBV preS1, considered to be the primary NTCP binding region of the viral L envelope protein, and showed that the chimeric HBV construct could infect marmoset cells more efficiently than wild-type HBV. This second part of the current work is novel and could be highly significant for the development of a more convenient primate model of HBV infection. However, a number of important technical caveats remain, which need to be addressed to strengthen the authors' conclusions.

Author reply: We thank this reviewer for their overall positive assessment of our study.

Major issues:

1. A major issue relates to how HBV infection was measured. How did the authors exclude the possibility that the HBsAg, HBeAg, HBcAg, HBV DNA and RNA detected in the infected cells may represent, partially or completely, input signals since all these materials were present in the inoculum? This is by no means a trivial issue given the high amounts of inoculum used and the low levels of replication reached in the current cell culture systems. This issue needs to be more vigorously addressed to clearly demonstrate a productive infection, which was inefficient at best in many of the experiments the authors report here.

Author reply: Following the suggestions, we performed additional experiments to further strengthen our conclusions including HBV DNA Southern blot, HBV RNA in situ hybridization, HBc immunohistochemistry, and stringent HBV cccDNA specific Hirt extraction and qPCR assays. In the Southern blot analysis of HBV DNA in the marmoset NTCP expressed HepG2 cells infected with HBV or our HBV/WMHBV preS1[1-48] chimeric virus, our results showed that after infection, different HBV DNA replication intermediate bands (rcDNA, dsDNA, ssDNA and other smeared signals indicating different lengths of negative single strands or double strand DNA with variety length of HBV positive strand) (see the revised **Fig. 5I**). More importantly, consistent with our other infection data in Fig. 6, the Southern blot signals in the HBV/WMHBV preS1[1-48] chimeric virus are stronger than in the HBV group.

We also quantified HBV cccDNA by qPCR strictly following the most updated HBV cccDNA qPCR quantification protocol (Allweiss et al. *Gut*, 2023; PMID: 36707234; Xia et al. *Methods in Molecular Biology*, 2017; PMID: 27975308) by Hirt DNA extraction, T5 exonuclease digestion and a cccDNA specific qPCR program. Our cccDNA qPCR readily detects cccDNA both in HBV and our HBV/WMHBV preS1[1-48] chimeric virus infection groups. More importantly, the cccDNA level in the HBV/WMHBV preS1[1-48] chimeric virus infection group is higher than that in the HBV infection group (revised **Fig. 5H**).

We also performed HBV RNA in situ hybridization (ISH) and HBc immunohistochemistry (IHC) by using the Formalin Fixed Paraffin Embedded (FFPE) samples to confirm the infection (revised **Fig. 5J**). since we believe HBV RNA is another HBV product that can only be detected after successful de novo infection.

2. The authors showed some results measuring cccDNA, which should not be present in the inoculum, by PCR. However, PCR-based methods for cccDNA detection and quantification are well known to be complicated by the fact that the HBV rcDNA in the inoculum can be detected also albeit at lower efficiency. Since the input rcDNA and that produced in the cell following productive infection far outnumber cccDNA, PCR detection of cccDNA under these conditions have to be carefully controlled. The author did not appear to include any methods on how they measured cccDNA. They should report the PCR method details and show how their methods detected only cccDNA but not rcDNA. Alternatively, they can use Southern blot analysis, which remains the gold standard for cccDNA detection.

Author reply: Following the suggestions, we performed Southern blot and HBV cccDNA specific Hirt extraction and qPCR by the most updated cccDNA protocol and the detailed methods were added in the material and methods part (lines 580-603). Additionally, we performed HBV RNA in situ hybridization (ISH) which allowed us to detect de novo transcribed viral RNA following infection. For more details and data illustrations please see our detailed answers to the question 1.

3. The apparently discordant results among the infection assays, e.g., HBsAg and HBeAg secretion (Fig. 5A and supplem Fig. 6D), intracellular HBcAg staining (Fig. 4G, supplem Fig. 7B and 8A), cccDNA PCR (supplem Fig. 6F), add to the concern as to whether productive infection was indeed established and whether it was accurately measured. These should be clarified with additional experiments suggested above.

Author reply: We apologize that the way we had originally presented these data appears to have led to some misunderstanding. To clarify, our original supplementary Fig.6D was data from transfection experiments to characterize the antigen expression and viral replication of HBV, WMHBV and the chimeric HBV/WMHBV preS1[1-48], not infection. From this different virus characterization data, we realized that the HBsAg secretion and HBV DNA replication levels among these three viruses were comparable, while WMHBV yielded higher HBeAg expression. This discrepancy can be explained by different activity at the basal core promoter or core promoter. HBsAg and HBeAg are commonly used markers for HBV infection, yet HBsAg is more frequently used because of its higher sensitivity. Since the infectivity of the HepG2-marmoset-NTCP cells lines is not very high, we did not robustly detect HBeAg in the supernatants. Primary marmoset hepatocytes were more susceptible and consequently, we detected significant levels of HBeAg. This is consistent with observations in primary hepatocytes and iPS-derived hepatocyte-like cells We have commented about this in the discussion section in the revised manuscript (lines 274-283).

Additional experiments suggested above were performed to strengthen our conclusions. Please see our detailed answers above.

Minor

issues:

4. Fig. 1. More clear and consistent labels are needed to match the host species in panel A vs. B-D so that it will be clear to the reader what are the exact NTCP sequences from OWM, SqM, Mammot, and Prosimian species used in B-D?

Author reply: Following the suggestion, the species in panel B-D were labeled in panel A accordingly and the figure was shown in the updated **Fig. 1A** and the revised figure legends (line 1015).

5. Fig. 2. It is clear that residues 84-87 of NTCP, while not critical for preS1 binding, is critical for

infection, a conclusion also reached by others previously. Indeed, mutants in this region enhanced infection (Fig. 2E and supplem Fig. 5B) but decreased preS1 binding (supplem Fig. 5A). Based on these results and the recently published structural data of NTCP and NTCP-preS1 complex, can the authors provide an plausible explanation(s) for these results.

Author reply: We thank the reviewer for their careful comments. In this study, we indeed found humanizing mutations on sqM-NTCP 84-87 region reduced preS1 binding ability and on the other hand can enhance HBV infection. Even though three papers were published nearly at the same time in *Nature* last year (Goutam et al. *Nature*, 2022; PMID: 35545671; Park et al. *Nature*, 2022; PMID: 35580630; Asami et al. *Nature*, 2022; PMID: 35580629), only Asami et al. reported on a cryo-EM structure of human NTCP in the presence of myristoylated preS1. However, as stated by the authors, their data quality was insufficient for model building. AA84-87 localize to an extracellular loop between TM2 and TM3, roughly 30 Å from residue 157 and one of the residues in this loop region affect bile acid uptake (Yan et al. *J Virol*, 2014; PMID: 24390325). HBV binding negatively correlates with bile acid uptake and HBV preS1 and bile acid share the same functional tunnel (Asami et al. *Nature*, 2022; PMID: 35580629). Thus, we think that the residues within 84-87 region are not important for HBV binding but rather later during HBV internalization. The HBV-NTCP interaction might cause conformational changes necessary for the initiation of endocytosis. We analyzed more carefully the NTCP structure and found that the humanizing mutations in sqM-NTCP 84-87 can increase the stabilization of the loop and lead the hydrophobicity more similar to the counterpart of hNTCP. We think the more stable status of NTCP structure in the TM2-3 loop may not benefit the binding but will help with viral entry. We added our comments in the revised manuscript (lines 136-138; 143-145; 250-255).

6. Fig. 4C. “HBsAg” should be labeled as WMHBsAg to avoid confusion.

Author reply:

Following the suggestion, the labels in all the figures were checked and revised accordingly (revised **Fig. 4C**, **4F** and **Fig. 6A**).

Reviewer #4 (Remarks to the Author):

This carefully designed study describes the generation of the simian tropic virus and the establishment of a marmoset animal model for HBV infection. There is a large need to develop a smaller NHP model of HBV infection since current murine models have limitations. By conducting a mutational analysis of the HBV receptor (sodium taurocholate co-transporting polypeptide, NTCP) orthologs from different primate species, the authors determined several NTCP residues critical for HBV entry into the host cell. Additionally, the authors established primary and stem-cell-derived marmoset hepatocyte cultures that supported infection of HBV or chimeric HBV harboring 48 amino acid residues from woolly monkey HBV. The study is well-designed, the manuscript is clearly written, and the results constitute a significant research accomplishment in the HBV field. However, structures from panel 3C need to be carefully inspected before the manuscript is accepted for publication.

Author reply: We thank this reviewer for their overall very positive assessment of our study.

Major comments:

1. It is not clear why the mutagenesis analysis in the 84-87 regions was not performed for OWM NTCPs. In particular, it would be interesting to see the effect of mutations in residues 84 and 86.

Author reply:

We thank this reviewer for their comment. Following the reviewer's suggestions, we introduced humanizing mutations in the 84-87 regions of OWM NTCP for HBV infection. Furthermore, double mutations in the 84-87 and 157-165 regions of OWM NTCP were also made to compare the HBV infectivity. Our infection data are in line with our previous results (**Fig. 2E; Supplementary Fig. 5B&5E**), i.e. that the single R158G mutation can confer OWM NTCP HBV infection and humanizing the whole counterpart of 157-165 increased HBV infection (the main contribution coming from the P165L mutation based on the data in this study in **Fig. 2E, Fig. 3A&B and Supplementary Fig. 5B**). However, as seen with the wild type OWM NTCP, Q84R and N86K mutations did not enhance HBV infection when introduced into in OWM NTCP (**Supplementary Fig. 6**). Furthermore, the double mutation of Q84R and N86K plus humanization of 157-165 did not increase HBV infectivity when compared to humanization of 157-165 region of OWM NTCP (**Supplementary Fig. 6**), suggesting that the Q84R and N86K mutations do not further enhance the ability of OWM NTCP to support HBV infection.

We undertook considerable efforts and performed extensive mutagenesis analyses to determine which amino acids restrict HBV infection across NTCP orthologues from different species. By comparing NWM and OWM NTCP, we found that OWM NTCP neither supports HBV binding, nor infection. However, NWM NTCP (SqM) supports HBV binding but not entry. By introducing the R158G gain-of-function mutation in OWM NTCP) and the reciprocal G158R loss-of-function mutation in SqM NTCP, we demonstrate that G158 is the determinant for HBV binding (**Fig. 2C&2D; Supplementary Fig. 4C**). Since the binding is the prerequisite for infection and R158G & P165L double mutations in OWM NTCP confer equivalent or even higher HBV infection ability as compared to human NTCP (**Fig. 2E; Supplementary Fig. 5B**), we reasoned that the main issue on OWM NTCP is its limited ability to support HBV binding. The differences between OWM NTCP and human NTCP in 84-87 region (only Q84 and N86 are different with human NTCP) do not seem to be important for HBV infection. More importantly, we found amino acids 84-87 are important for HBV entry in NWM NTCP. By introducing the humanizing Q84R mutation and even more so the K87N mutation in SqM NTCP, we show that these single mutations can enable HBV infection (**Fig. 2E**). Because OWM NTCP naturally harbors an asparagine in position 87, the amino acids in 84-87 region in OWM NTCP are not responsible for its inability to support HBV infection. Taken these points together, we did not perform an additional mutagenesis analysis in

the 84-87 regions for OWM NTCP in our original manuscript. We thank this reviewer for their comment as the newly added data further strengthen our findings and conclusions.

Additionally, have the authors performed mutagenesis experiments for marmoset NTCPs?

Author reply:

We thank this reviewer for raising this point. Following the reviewer's suggestion, we performed additional mutagenesis analysis on marmoset NTCP. Since both marmosets and SqMs belong to NWMs and based on our data in this study, the inability of NWM NTCP to support HBV infection can be mainly attributed to differences in the 84-87 region. We used SqM as a representative species for NWM and extensively mutagenized SqM NTCP which allowed us to determine that the Q84R mutation and even more so the K87N single mutation in SqM NTCP can enable HBV infection (**Fig. 2E**). On the other hand, marmoset NTCP naturally harbors an arginine residue in position 84 and thus we only introduced the K87N humanizing mutation in marmoset NTCP and performed HBV binding experiments. Our results are consistent with the equivalent mutation in SqM NTCP showing that the K87N mutant can reduce HBV binding. These new data are now included in the updated Supplementary **Fig. 5C&5D**.

2. In figure 3C, It appears as hNTCP and SqMWT structure alignments might have been switched. Amino acids labels look correct, but the amino acid side chains do not correspond with the amino acid labels.

Author reply:

We thank this reviewer for pointing out these inaccuracies. We re-analyzed the NTCP structure data and created revised visual representations of the 84-87 region. To highlight the important residues (position 84 and position 87) responsible for HBV infection that we found in this study, we labeled their side-chains in red in a stick model. Since the 84-87 region localizes to the loop region between the transmembrane (TM) 2 and TM 3(a) of the NTCP core domain, the structures of 84-87 region plus the surrounding TM2 and TM3 among different NTCP orthologs were also represented separately for comparison. However, the overall structures are very similar for this region and all their alignment distance (showed as RMSD score) to human NTCP is around 0.1 Å. The important differences mapping to the side chains of 84-87 amino acids may affect HBV's ability to interact with the receptor orthologue and/or can possibly affect putative spatial conformational changes that might be occurring during the viral entry process. The revised data are now shown in the updated **Fig. 3C** and **Supplementary Fig 3D**.

3. Again, in figure 3C, SqMK87N does not appear to have a side chain corresponding to asparagine at position 87. Please check all models from panel 3C.

Author reply:

The structure data were checked carefully and revised accordingly (updated **Fig. 3C**).

Minor comments:

1. Line 65, "mic" should be changed to "mice"

Corrected (line 67).

2. Throughout the text, "cyro-EM" should be changed to "cryo-EM" (lines 122, 861 in the main text; lines 88, 93 in the supplemental figures file)

We read through our manuscript carefully and revised this accordingly (lines 114, 125, 801, 806, 1030 in the revised manuscript and in the supplementary files).

3. Line 307, Can you please clarify whether H_μrelhuman-96 SACC-PMHs were used in this study?

From the experiment setup, it appears as SACC-PMHs were constructed from marmoset, not human hepatocytes

Thank you for pointing out this inaccuracy. All the self-assembling co-cultures used in this study were constructed from marmoset. We have corrected in the revised manuscript (line 334).

4. Line 850, Please add “pre S1” before “binding assay”

Added (line 1018).

5. Lines 863-868, panels C and D need separate descriptions.

Separate descriptions were made for the Figure legends in the revised manuscript (lines 1032-1039).

*6. Line 872, Figure 2E needs to have description for *****

Added (line 1043).

7. Lines 874-881, please specify whether any PDB models were used for structure alignment

The human NTCP structure models in Figure 3 were from the public PDB database (PDB ID: 7PQG and 7PQQ). The information was added in the figure legends correspondingly in the revised manuscript (lines 1046-1053).

8. Figure 1D, X-axis labels (6 and 8), need to be rotated 90° clockwise

Corrected (updated **Fig. 1D**).

9. In figure 3C, please remove hydrogen atoms from all models

The hydrogen atoms from all models were removed and the revised structure data were shown in the updated **Fig. 3C**.

REVIEWERS' COMMENTS

Reviewer #1 (Remarks to the Author):

The authors appropriately answered to all of the questions raised by the reviewer, with adding new experiments. The manuscript has been significantly improved and I support the publication of this paper.

Reviewer #2 (Remarks to the Author):

The authors have addressed all of our raised concerns adequately

Reviewer #3 (Remarks to the Author):

The revised manuscript addressed adequately my concerns in the previous review. I have no more major issues.

Reviewer #4 (Remarks to the Author):

The authors have responded appropriately to all my comments and suggestions. I have no further issues with this manuscript.

Rebuttal:

Reviewer #1:

The authors appropriately answered to all of the questions raised by the reviewer, with adding new experiments. The manuscript has been significantly improved and I support the publication of this paper.

Reviewer #2:

The authors have addressed all of our raised concerns adequately

Reviewer #3:

The revised manuscript addressed adequately my concerns in the previous review. I have no more major issues.

Reviewer #4:

The authors have responded appropriately to all my comments and suggestions. I have no further issues with this manuscript.

We are glad that we were able to address satisfactorily all points raised by the reviewers. We thank the reviewers for their thorough edits and suggestions which have helped to improve the manuscript.